# GRAPH-REFINED REPRESENTATION LEARNING FOR FEW-SHOT CLASSIFICATION VIA CLIP ADAPTATION

## ABSTRACT

Few-shot image classification remains a fundamental challenge, as learning transferable representations from only a handful of examples often fails to generalize to unseen concepts. Recent advances benefit pre-trained vision-language models such as CLIP, yet their inherent biases and limited task adaptability hinder robust performance. We propose a novel *graph-driven cache refinement framework* that improves CLIP's prior knowledge with task-specific representation learning while preserving its lightweight inference. The first stage, **Inductive Statistical Subspace Aggregation (ISSA)**, partitions each feature into subspaces, builds fully connected intra-sample graphs, and applies statistical aggregation to capture robust subspace-level dependencies. The second stage, **Feature Subspace Propagation (FSP)**, globally diffuses contextual signals across subspaces while preserving their individuality, resulting in enriched embeddings that drive cache-based retrieval models. In particular, this refinement branch is active only during training, producing enhanced cache keys while ensuring graph-free, efficient inference. Across multiple benchmarks, our method consistently outperforms state-of-the-art approaches, establishing new performance standards in few-shot learning while retaining computational efficiency. Source code will be released to support reproducibility and further research.

## 1 INTRODUCTION

Vision-Language Models (VLMs), such as CLIP (Radford et al., 2021), have redefined representation learning by aligning images and text on scale (Jia et al., 2021; Yao et al., 2021). Trained with a contrastive objective on large image-text pairs, they yield powerful multimodal embeddings that transfer remarkably well across tasks. In this setting, *few-shot classification* has emerged as a critical benchmark for testing adaptation under limited supervision. It is usually studied in two settings: *inductive*, which independently predicts each test image and includes prompt-, adapter-, and cache-based methods (Zhou et al., 2022b; Gao et al., 2024; Zhang et al., 2022); and *transductive*, which exploits batch-level statistics via label propagation or manifold regularization (Ziko et al., 2020; Boudiaf et al., 2020; Zhu & Koniusz, 2023). CLIP has proven effective in aligning both image features with textual prompts such as "a photo of a `[CLASS]`". However, despite its strong zero-shot performance, it struggles in fine-grained or distribution-shifted scenarios and remains highly sensitive to prompt design (Liu et al., 2023; Brown et al., 2020). These limitations have motivated a growing set of few-shot adaptation strategies, particularly in the inductive regime, aiming to refine CLIP's priors into task-specific representations.

Within the inductive setting, three dominant strategies have emerged. **Prompt-learning** optimizes the text branch by replacing hand-crafted prompts with a few learned continuous tokens. This often outperforms manual prompts with as few as one or two shots (Zhou et al., 2022b; Chen et al., 2023; Khattak et al., 2023; Zhu et al., 2023). **Adapter-based** methods add lightweight trainable layers to image and/or text encoders, achieving competitive or superior performance to prompt tuning with a little extra computation (Gao et al., 2024). In contrast, **training-free retrieval** avoids any parameter updates by constructing a cache of few-shot features and combining $k$-NN scores with CLIP's zero-shot logits (Zhang et al., 2022). This reveals a clear trade-off: prompts and adapters adapt CLIP more effectively but incur training overhead, while training-free retrieval is efficient and preserves zero-shot generality but fails to exploit structural relationships within CLIP's embedding space. We address this by partitioning each CLIP feature into subspaces and refining them with a lightweight

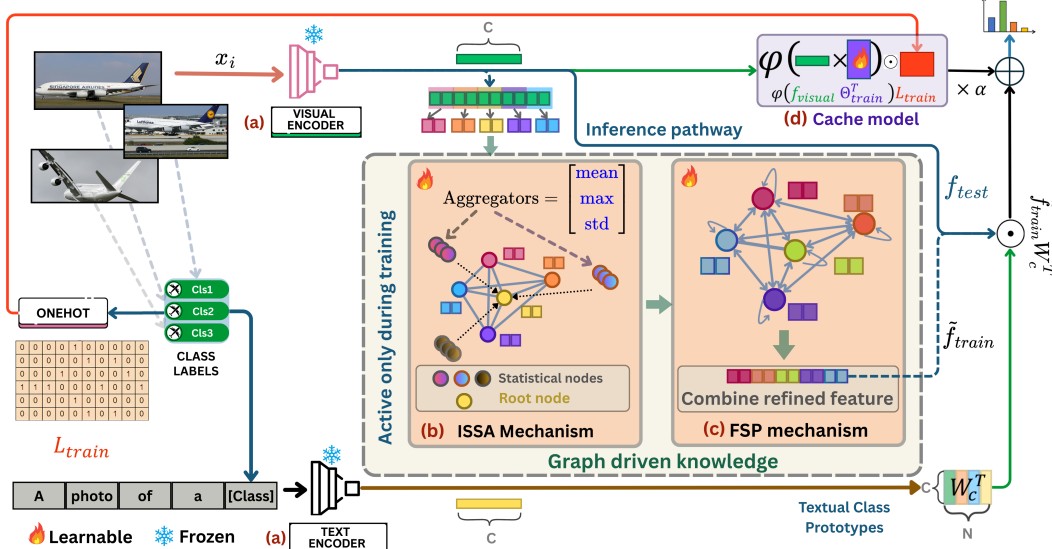

Figure 1: Overview of our graph-driven representation learning for CLIP adaptation in few-shot classification. (a) The CLIP encoder extracts visual features, partitioned into $|S|$ subspace nodes. (b) ISSA aggregates neighborhood statistics over the fully connected subspace graph. (c) FSP propagates global context while preserving node identity via residual connections. (d) The refined embeddings are fused into a key–value cache, enabling efficient few-shot classification.

graph during training. The resulting enriched subspaces are then used to refine cache keys, yielding graph-free inference that is both efficient and significantly more accurate.

**Motivation:** In few-shot learning, CLIP is typically frozen and produces a single global embedding per image. This embedding often exhibits **few-shot bias**: it heavily relies on strong pre-training signals, such as, dominant texture, color, or prompt-related directions, while suppressing the fine-grained or task-specific cues needed for new dataset. Standard cache-based methods methods treat this global embedding as a **monolithic vector**, making it hard to adjust or reweight its components with limited supervision. Recent analyses in metric learning (Li et al., 2023) and self-supervised (Wang et al., 2021) representation learning suggest that deep features naturally organize into **semantic channel groups**, and that modeling their interactions can improve representation quality. Motivated by this, we partition each CLIP feature into subspaces that can exchange information and surface complementary evidence inaccessible to a single global vector. We refine these subspaces with a lightweight graph composed of two steps: (i) a *statistical aggregation* that models each subspace as a node and uses message passing to explicitly encode dependencies and adaptively aggregates complementary evidence (Chen et al., 2024; Jiao et al., 2022), followed by (ii) a *propagation step* that diffuses global context across subspaces while preserving their individual identity (Figure 1). Without edges in the graph, refinement becomes equivalent to independent MLP updates, discarding relational cues. We adopt a simple, contiguous partitioning of CLIP embeddings. This is not due to the indices being semantically ordered, but because dense aggregation and propagation across subspaces allow the model to discover meaningful relations during training. More importantly, our graph refinement is applied only in the training phase to enrich the feature representation that refines cache keys, ensuring that inference remains *graph-free*, *efficient*, and *scalable*. Empirically, we find that removing inter-subspace edges (relationships) consistently degrades performance, while refined caches exhibit stronger subspace co-activation and deliver consistent accuracy gains on standard few-shot benchmarks.

## 2 RELATED WORKS

**Vision-Language Model for Few-Shot Learning:** Few-shot adaptation of VLMs has converged on two parameter-efficient directions. **Prompt learning** freezes CLIP (Radford et al., 2021) and optimizes a small set of continuous tokens: CoOp (Zhou et al., 2022b) learns class-agnostic text prompts, CoCoOp extends them with instance-conditioned context (Zhou et al., 2022a), PLOT++ formulates prompt search as optimal transport (Chen et al., 2023), KgCoOp injects knowledge-guided constraints (Yao et al., 2023), MaPLe links image- and text-side prompts (Khattak et al., 2023), and

ProGrad regularizes updates via gradient alignment (Zhu et al., 2023). These methods excel in an extreme few-shot setting (1–4), but saturate as the shots increase. **Adapter-based** tuning instead inserts lightweight modules into the vision stream: CLIP-Adapter adds residual bottlenecks (Gao et al., 2024), and TaskRes learns task-specific side branches (Yu et al., 2023). Adapters generally require short training, but preserve zero-shot priors better than full fine-tuning. Overall, prompts and adapters expose a trade-off: a stronger task comes at the cost of additional parameters. Recent works such as Ta-Adapter (Zhang et al., 2024) and CCA (Jiang et al., 2025) explore *task-aware and causally grounded adapters* to push CLIP beyond lightweight tuning. Ta-Adapter injects task-aware prompts into CLIP's encoders for deeper adaptation but introduces additional trainable modules that must remain active during inference, while CCA disentangles CLIP features via Independent Component Analysis (ICA) and adds further cross-modal alignment modules: both requiring extra inference-time components. Our method is complementary to these trends: we preserve frozen CLIP encoders and *distill* graph-refined knowledge entirely into the cache keys, keeping test-time complexity identical to standard cache models.

**Cache Model:** Cache-driven methods (Khandelwal et al., 2020; Orhan, 2018; Zhang et al., 2022) offer a lightweight alternative to fine-tuning for few-shot adaptation. Instead of updating network parameters, they store training features as key–value pairs, where keys are embeddings and values are class labels. In inference, a test feature retrieves the most similar keys, allowing efficient prediction (Khandelwal et al., 2020; Orhan, 2018). Early cache models such as kNN-LMs (Khandelwal et al., 2020) and Matching Networks (Vinyals et al., 2016) focused on unimodal features and often suffered from storage overhead or noisy retrieval. Tip-Adapter (Zhang et al., 2022) extended this idea by encoding few-shot images as cache keys and combining retrieval scores with zero-shot CLIP logits through a residual connection. Its fine-tuned variant Tip-Adapter-F (Zhang et al., 2022) further updates the keys via lightweight SGD, improving accuracy with minimal training. Using CLIP's multimodal pre-training, Tip-Adapter retains robustness under distribution shift while remaining efficient. Our approach builds on this retrieval backbone but redefines how keys are constructed: each CLIP embedding is partitioned into statistical subspaces and refined once with a lightweight graph during cache learning. The enriched keys yield higher retrieval accuracy at test time, while inference remains standalone, fast, and entirely graph-free.

**Few-shot Graph Learning:** Graph-based propagation shares information between the sparse support sets of few-shot tasks. Early metric learners such as Matching Networks (Vinyals et al., 2016), Prototypical Networks (Snell et al., 2017), and Relation Networks (Sung et al., 2018) can be viewed as performing one round of message passing on a similarity graph linking support and query embeddings. GNNs make this propagation explicit and enable multi-hop reasoning: GCNs (Kipf & Welling, 2017) rely on fixed graphs and remain transductive, while GraphSAGE (Hamilton et al., 2017) learns inductive aggregation functions that generalize to unseen nodes. Few-Shot Learning with GNN (Garcia & Bruna, 2018) extends this idea with learnable edge functions and stacked propagation, effectively unifying previous metric methods. Subsequent work refined this direction with task-specific affinity kernels (Liu et al., 2019), Laplacian-regularized clustering (Ziko et al., 2020), and meta-learned propagation operators (Kim et al., 2019). Our approach differs from these designs: rather than propagating over support–query graphs, we decompose each frozen CLIP embedding into statistical subspaces and refine them via a lightweight graph. A single global propagation step diffuses contextual cues across subspaces, resulting in higher-quality cache keys for inductive few-shot adaptation while keeping inference graph-free.

## 3 PROPOSED METHOD

**Problem Formulation:** We tackle the few-shot classification via semi-parametric adaptation. Consider a support set $\mathcal{D}_{\text{train}} = \{(x_i, y_i)\}_{i=1}^{M}$ consisting of $M = N \cdot K$ images, representing $N$ classes with $K$ samples per class, and an unlabeled test set $\mathcal{D}_{\text{test}} = \{x_j\}$ drawn from the same label space. Our goal is to leverage a pre-trained VLM $\mathcal{F}$ (e.g., CLIP (Radford et al., 2021)) and adapt it efficiently to achieve effective inference in $\mathcal{D}_{\text{test}}$ using only the set $\mathcal{D}_{\text{train}}$. We introduce a learnable cache model $\mathcal{C} = (\Theta_{\text{train}}, L_{\text{train}})$, where $\Theta_{\text{train}} \in \mathbb{R}^{M \times C}$ initialized with feature embeddings $f_i$ extracted from $\mathcal{D}_{\text{train}}$ via $\mathcal{F}$, and $L_{\text{train}} \in \mathbb{R}^{M \times N}$ contains the corresponding one-hot class labels. To enhance the quality of the representation, we incorporate graph-driven representation learning to enrich $f_i$ during training (Figure 1). This enriched $f_i$ refines the cache model $\mathcal{C}$ during training.

**Few-Shot Cache Construction:** We construct a key-value cache $\mathcal{C}$ from the set $\mathcal{D}_{train}$ based on the concept of the Tip-Adapter (Zhang et al., 2022). Each image $x_i \in \mathcal{D}_{train}$ is passed through

the CLIP's frozen visual encoder $\mathcal{F}_{VisEnc} \in \mathcal{F}$ to obtain a L2-normalized $C$-dimensional vector as $f_i = \mathcal{F}_{VisEnc}(x_i) \in \mathbb{R}^{1 \times C}$. This features set $\phi = \{f_i\}_{i=1}^M$ computed from $\mathcal{D}_{train}$ are stacked into the matrix $\Theta_{\text{train}} \in \mathbb{R}^{M \times C}$. Consequently, each label $y_i \in 1, \ldots, N$ is one-hot encoded as $l_i \in \mathbb{R}^N$ and stacked to form $L_{\text{train}} \in \mathbb{R}^{M \times N}$. This results in the construction of our key-value cache model $\mathcal{C} = (\Theta_{\text{train}}, L_{\text{train}})$. To enhance cache expressiveness, we adopt a learnable cache where the key matrix $\Theta_{\text{train}}$ is refined during fine-tuning with graph-enhanced representations, while the value matrix $L_{\text{train}}$ remains fixed. In inference, prediction reduces to efficient retrieval: the test query feature is matched against cached keys.

**Graph-based Subspace Interaction:** A visual feature $f_{\text{train}} \in \phi$ is divided into $S$ non-overlapping subspaces $\{f_{\text{train}}^j\}_{j=1}^{\mathcal{S}}$, where each $f_{\text{train}}^j \in \mathbb{R}^{1 \times C/\mathcal{S}}$ denotes a feature subspace. These subspaces form nodes in a fully connected undirected graph $G = (V, E)$, with $|V| = \mathcal{S}$ and $|E| = \binom{\mathcal{S}}{2} = \mathcal{S}(\mathcal{S} - 1)/2$. The graph $G$ enables each subspace to interact with all others, capturing intra-feature relationships such as semantic co-activations and contextual dependencies. The interactions among subspaces are processed in two stages via our proposed GNN, resulting in updated node embeddings that are concatenated into an enhanced representation $\tilde{f}_{\text{train}} \in \mathbb{R}^{1 \times C}$. This subspace-aware representation enriches cache keys with higher-order semantics during training, while inference relies solely on the refined cache for prediction.

**Graph-driven Representation Learning for Knowledge Incorporation in Cache:** Our design builds on the inductive aggregation philosophy of Hamilton et al. (Hamilton et al., 2017), which demonstrated that learning transferable aggregation functions for node neighborhoods enables generalization to unseen graphs and nodes. Inspired by this, we advance it to our subspace graphs using the ISSA (**Inductive Statistical Subspace Aggregation**) module. Unlike neighbor sampling schemes used in large-scale GNNs, ISSA operates on fully connected subspace graphs and applies complementary statistical functions such as mean, max, and standard deviation over each subspace neighborhoods. This dense, all-to-all interaction captures fine-grained relational dependencies and discovers robust feature representations that transfer across diverse few-shot tasks.

To further enhance global coherence, we introduce **Feature Subspace Propagation** (FSP), inspired by personalized propagation frameworks such as APPNP (Klicpera et al., 2019). FSP diffuses contextual information across the entire subspace graph to enhance the semantic consistency and robustness of the representations by iteratively mixing node features while retaining their original identity through controlled residual connections. Together, ISSA and FSP form a lightweight refinement pipeline: ISSA provides strong local relational cues, while FSP enforces semantic consistency and robustness across subspaces. Formally, let $v$ denote a subspace node with embedding $\mathbf{h}_v^{(k-1)}$ at layer $k - 1$, and let $\mathcal{N}(v)$ denote its neighbors. $\mathbf{h}_v^{(0)}$ is the original subspace features i.e., $\mathbf{h}_v^{(0)} \in \{f_{\text{train}}^j\}_{j=1}^{\mathcal{S}}$. ISSA applies three statistical aggregation functions (mean, max, and standard deviation) to summarize the neighborhood information, followed by a self-attention mechanism to adaptively weight these aggregated signals:

$$\text{AGG}^{(k-1)}(\{\mathbf{h}_u^{(k-1)}\}_{u \in \mathcal{N}(v)}) = \frac{1}{|\psi|} \sum_\psi \left( \text{softmax}\left( \frac{(\mathbf{H}_u \mathbf{W}_Q)(\mathbf{H}_u \mathbf{W}_K)^\top}{\sqrt{F}} \right) \mathbf{H}_u \mathbf{W}_V \right) \mathbf{W}_O \quad (1)$$

where $\psi = \{\text{mean}(\cdot), \text{max}(\cdot), \text{std}(\cdot)\}$, and $\mathbf{H}_u = \left[ \mathbf{W}_{\text{mean}} \mathbf{h}_{\text{mean}}^{(k-1)}, \; \mathbf{W}_{\text{max}} \mathbf{h}_{\text{max}}^{(k-1)}, \; \mathbf{W}_{\text{std}} \mathbf{h}_{\text{std}}^{(k-1)} \right] \in \mathbb{R}^{3 \times F}$ with $\mathbf{h}_{\text{mean}}^{(k-1)}, \mathbf{h}_{\text{max}}^{(k-1)}, \mathbf{h}_{\text{std}}^{(k-1)}$ computed over $\{\mathbf{h}_u^{(k-1)}\}_{u \in \mathcal{N}(v)}$, and $\mathbf{W}_{\text{mean}}, \mathbf{W}_{\text{max}}, \mathbf{W}_{\text{std}} \in \mathbb{R}^{F \times F}$ denoting learnable projections for each statistical node. Here, $F = C/\mathcal{S}$ is the feature dimension of each subspace, and $\mathbf{W}_Q, \mathbf{W}_K, \mathbf{W}_V, \mathbf{W}_O \in \mathbb{R}^{F \times F}$ are query, key, value, and output projections used in the self-attention mechanism. Here, the neighborhood statistics (mean, max, std) are stacked as rows of a $3 \times F$ matrix in $H_u$, where each row serves as an individual "statistical node" rather than collapsing into a single $1 \times 3F$ vector. We then apply our shared self-attention mechanism (with $\mathbf{W}_Q$, $\mathbf{W}_K$, $\mathbf{W}_V$, and $\mathbf{W}_O$) over these nodes and sum the self-attended outputs to adaptively reweight and fuse their complementary information, producing a single refined $1 \times F$ representation for each subspace. This aggregated neighborhood context for node $v$ is then passed through a learnable linear transformation $\mathbf{W}_{\text{u}}^{(k-1)}$ and combined with a linearly transformed version of the node's own embedding $\mathbf{W}_{\text{self}}^{(k-1)} \mathbf{h}_v^{(k-1)}$. The result is then passed through a nonlinearity $\sigma$

(e.g., ReLU), which results in the ISSA update at the layer $k$:

$$\mathbf{h}_v^{(k)} = \sigma\Big(\underbrace{\mathbf{W}_{\text{self}}^{(k-1)}\mathbf{h}_v^{(k-1)}}_{\text{self-feature}} + \underbrace{\mathbf{W}_{\text{u}}^{(k-1)}\text{AGG}^{(k)}\big(\{(\mathbf{h}_u^{(k-1)}) : u \in \mathcal{N}(v)\}\big)}_{\text{neighbor context}} + \mathbf{b}_u^{(k-1)}\Big) \qquad (2)$$

The subspace-aware local representations produced by ISSA are further refined through FSP, which injects high-order semantic context to produce enriched feature representations and higher-quality cache keys. Let $\mathbf{H}^{(k)} \in \mathbb{R}^{\mathcal{S} \times F}$ denote the matrix of subspace representation by ISSA, where each row $\mathbf{h}_v^{(k)}$ corresponds to node $v$. FSP performs $T$ propagation steps and incorporates a teleport mechanism to preserve the original node's features. At iteration $t = 1, \ldots, T$, the nodes' representations are updated as:

$$\mathbf{H}^{(k)(0)} = \mathbf{H}^{(k)}; \quad \mathbf{H}^{(k)(t)} = \gamma \mathbf{H}^{(k)(0)} + (1-\gamma)\tilde{\mathbf{A}}\mathbf{H}^{(k)(t-1)}; \quad \mathbf{H}^{(k+1)} = \mathbf{H}^{(k)(T)}, \qquad (3)$$

where $\gamma \in [0,1]$ controls the strength (teleport probability) of the residual connections compared to the initial subspace embedding. The matrix $\tilde{\mathbf{A}} = \hat{\mathbf{D}}^{-1/2}\hat{\mathbf{A}}\hat{\mathbf{D}}^{-1/2}$ is the symmetrically normalized adjacency matrix with self-loops, where $\hat{\mathbf{A}} = \mathbf{A} + \mathbf{I}$ and $\hat{\mathbf{D}}$ is the diagonal matrix. Eq. (3) combines two key components: (1) a *reset term* $\gamma \mathbf{H}^{(k)(0)}$ that preserves the initial subspace representations, and (2) a *propagation term* $(1-\gamma)\tilde{\mathbf{A}}\mathbf{H}^{(k)(t-1)}$ that iteratively diffuses contextual information across nodes. In $T$ iterations, this process injects higher-order semantic dependencies into each subspace, resulting in globally enriched representations. The final output $\mathbf{H}^{(k+1)} \in \mathbb{R}^{\mathcal{S} \times F}$ contains refined subspace representations $\mathbf{H} = \{\mathbf{h}_{i,:}\}_{i=1}^{\mathcal{S}}$, with $\mathbf{h}_{i,:} \in \mathbb{R}^{1 \times (C/\mathcal{S})}$. These are then concatenated to form the enhanced representation $\tilde{f}_{\text{train}} = \bigoplus_{i=1}^{\mathcal{S}} \mathbf{h}_{i,:} \in \mathbb{R}^{1 \times C}$, where $\bigoplus$ denotes concatenation. It serves as an enriched representation that encodes relational cues for cache adaptation.

Following the Tip-Adapter (Zhang et al., 2022), we represent the cache as a set of key–value pairs: the keys $\Theta_{\text{train}} \in \mathbb{R}^{M \times C}$ are initialized with the L2-normalized CLIP visual features from the support set $\mathcal{D}_{\text{train}}$, and updated via gradient during training, while the values $L_{\text{train}} \in \mathbb{R}^{M \times N}$ are fixed one-hot label vectors. To generate predictions, we combine two complementary streams: (i) *cache-based retrieval:* the query feature $f_{\text{train}}$ is matched against cache keys $\Theta_{\text{train}}$ using cosine similarity, producing affinity weights that retrieve class information from $L_{\text{train}}$. (ii) *CLIP-based classification:* the refined embedding $\tilde{f}_{\text{train}}$ is projected onto CLIP's pre-trained textual prototypes $W_c \in \mathbb{R}^{N \times C}$. The final logits combine both streams:

$$\text{logits} = \alpha\varphi(f_{\text{train}}\Theta_{\text{train}}^\top)L_{\text{train}} + \tilde{f}_{\text{train}}W_c^\top, \qquad (4)$$

where $\alpha \in \mathbb{R}^+$ is a balancing coefficient and $\varphi(x) = \exp(-\beta(1-x))$ is a temperature-controlled ($\beta$) affinity function that computes similarity weights between the query and the keys. The larger $\beta$ values sharpen the similarity weighting, amplifying closer matches. Since both $f_{\text{train}}$ and $\Theta_{\text{train}}$ are L2-normalized, the similarity reduces to cosine distance. The corresponding affinity matrix is:

$$A = \exp\big(-\beta\big(1 - f_{\text{train}}\Theta_{\text{train}}^\top\big)\big), \qquad (5)$$

which, when multiplied with the fixed label matrix $L_{\text{train}}$, yields the cache-based prediction $AL_{\text{train}} \in \mathbb{R}^{1 \times N}$, representing the first term in Eq. (4) as the few-shot prediction component based on the learned cache. During training, the cache keys $\Theta_{\text{train}}$ and the graph branch are jointly optimized. The training logits are formed as a residual sum of the *first term*: $z_{\text{cache}}$ and the *second term*: $z_{\text{CLIP}}(\tilde{f}_{\text{train}})$ in Eq. (4), corresponding to (i) a cache stream using $\Theta_{\text{train}}$ and (ii) a graph-refined CLIP stream based on $\tilde{f}_{\text{train}}$. A cross-entropy loss $\mathcal{L}$ on this combined logits yields the gradient with respect to the keys as $\frac{\partial \mathcal{L}}{\partial \Theta_{\text{train}}} = \frac{\partial \mathcal{L}}{\partial \text{logits}} \cdot \frac{\partial z_{\text{cache}}}{\partial \Theta_{\text{train}}}$, where $\frac{\partial \mathcal{L}}{\partial \text{logits}}$ depends on the sum $z_{\text{cache}} + z_{\text{CLIP}}(\tilde{f}_{\text{train}})$. As the graph-refined CLIP branch becomes more predictive, it reshapes $\frac{\partial \mathcal{L}}{\partial \text{logits}}$ and thus indirectly guides the cache keys to align with the decision boundary induced by $\tilde{f}_{\text{train}}$. This is analogous to a residual knowledge-distillation setup where a strong teacher branch shapes the gradients on a student branch without explicit parameter sharing. The CLIP encoders remain frozen throughout; after training, the graph parameters are discarded and only the refined keys are used for retrieval.

**Final Prediction and Inference Pipeline:** At test time, inference relies solely on the refined cache learned during training without using our graph-driven subspace modeling. Given a test image, its feature $\mathbf{f}_{\text{test}} \in \mathbb{R}^{1 \times C}$ is extracted using the frozen CLIP visual encoder, following the same

Table 1: Few-shot classification accuracy (%) across 11 benchmark datasets for various state-of-the-art (SOTA) adapter-based approaches. Best are shown in **Bold**.

| Shots | Method | Venue | ImageNet | SUN | Aircraft | EuroSAT | Cars | Food | Pets | Flowers102 | Caltech | DTD | UCF101 | Average |
|---|---|---|---|---|---|---|---|---|---|---|---|---|---|---|
| 0 | CLIP [Radford et al., 2021] | ICML'22 | 60.3 | 58.5 | 17.1 | 37.5 | 55.7 | 77.3 | 86.1 | 66.0 | 85.9 | 42.2 | 61.5 | 59.2 |
| 1 | TIP-Adapter-F [Zhang et al., 2022] | ECCV'22 | 61.1 | 62.5 | 20.2 | 59.5 | 58.9 | 77.5 | 87.0 | 80.0 | 89.3 | 49.6 | 64.9 | 64.6 |
|  | TaskRes [Yu et al., 2023] | CVPR'23 | 61.9 | 62.3 | 21.4 | 61.7 | 59.1 | 74.0 | 83.6 | 79.2 | 88.8 | 50.2 | 64.8 | 64.3 |
|  | GraphAdapter [Li et al., 2024] | NeurIPs'23 | 61.5 | 61.9 | 20.9 | 63.3 | 59.7 | 75.4 | 84.4 | 80.0 | 88.9 | 51.8 | 64.9 | 64.8 |
|  | CLIP-Adapter [Gao et al., 2024] | IJCV'24 | 61.2 | 61.3 | 17.5 | 61.4 | 55.1 | 76.8 | 86.0 | 73.5 | 88.6 | 45.8 | 62.2 | 62.7 |
|  | CLAP [Silva-Rodriguez et al., 2024] | CVPR'24 | 58.5 | 61.1 | 20.6 | 59.2 | 56.3 | 73.0 | 83.6 | 79.9 | 88.4 | 47.5 | 62.5 | 62.8 |
|  | CAA [Jiang et al., 2025] | ICCV'25 | 61.5 | 63.8 | 22.5 | 68.0 | 60.0 | 77.8 | 86.9 | 81.0 | 89.9 | 51.0 | 66.3 | 66.3 |
|  | Proposed | - | 69.2 | 65.6 | 28.1 | 63.7 | 64.5 | 86.0 | 89.1 | 83.2 | 94.0 | 52.4 | 71.7 | **69.8** |
| 2 | TIP-Adapter-F [Zhang et al., 2022] | ECCV'22 | 61.7 | 63.6 | 23.2 | 66.1 | 61.5 | 77.8 | 87.0 | 82.3 | 89.7 | 53.7 | 66.4 | 66.6 |
|  | TaskRes [Yu et al., 2023] | CVPR'23 | 61.9 | 64.9 | 24.1 | 65.8 | 63.7 | 75.2 | 84.6 | 86.6 | 90.3 | 55.1 | 70.0 | 67.5 |
|  | GraphAdapter [Li et al., 2024] | NeurIPs'23 | 62.3 | 64.6 | 23.8 | 67.3 | 63.2 | 76.3 | 86.3 | 85.6 | 90.2 | 55.7 | 69.5 | 67.7 |
|  | CLIP-Adapter [Gao et al., 2024] | IJCV'24 | 61.5 | 63.3 | 20.1 | 63.9 | 58.7 | 77.2 | 86.7 | 81.6 | 89.4 | 51.5 | 67.1 | 65.5 |
|  | CLAP [Silva-Rodriguez et al., 2024] | CVPR'24 | 58.5 | 63.3 | 23.2 | 65.6 | 61.4 | 74.9 | 84.9 | 84.2 | 89.8 | 53.0 | 67.8 | 66.1 |
|  | CAA [Jiang et al., 2025] | ICCV'25 | 62.1 | 66.3 | 25.0 | 70.0 | 64.0 | 77.9 | 87.9 | 88.0 | 91.0 | 55.0 | 69.3 | 68.9 |
|  | Proposed | - | 69.7 | 67.9 | 31.2 | 67.4 | 64.5 | 86.1 | 90.0 | 88.7 | 94.5 | 55.3 | 75.5 | **71.9** |
| 4 | TIP-Adapter-F [Zhang et al., 2022] | ECCV'22 | 62.5 | 66.2 | 25.8 | 74.1 | 64.6 | 78.2 | 87.5 | 88.8 | 90.6 | 57.4 | 70.6 | 69.7 |
|  | TaskRes [Yu et al., 2023] | CVPR'23 | 63.6 | 67.3 | 25.7 | 73.8 | 67.4 | 76.1 | 86.3 | 90.2 | 91.0 | 60.7 | 70.9 | 70.3 |
|  | GraphAdapter [Li et al., 2024] | NeurIPs'23 | 63.1 | 66.7 | 27.0 | 75.2 | 66.5 | 76.8 | 86.6 | 89.9 | 91.0 | 59.6 | 71.5 | 70.3 |
|  | CLIP-Adapter [Gao et al., 2024] | IJCV'24 | 62.7 | 66.0 | 22.6 | 73.4 | 62.4 | 77.9 | 87.5 | 87.2 | 90.0 | 56.9 | 69.1 | 68.7 |
|  | CLAP [Silva-Rodriguez et al., 2024] | CVPR'24 | 60.7 | 65.9 | 25.6 | 73.1 | 65.5 | 75.9 | 86.5 | 87.6 | 90.6 | 58.8 | 69.8 | 69.1 |
|  | CAA [Jiang et al., 2025] | ICCV'25 | 63.3 | 69.0 | 29.0 | 80.1 | 68.0 | 78.2 | 88.1 | 91.1 | 92.0 | 63.0 | 72.1 | 72.2 |
|  | Proposed | - | 70.5 | 71.5 | 36.0 | 78.1 | 65.5 | 86.3 | 89.7 | 95.6 | 95.7 | 62.3 | 80.2 | **75.6** |
| 8 | TIP-Adapter-F [Zhang et al., 2022] | ECCV'22 | 64.0 | 68.9 | 30.2 | 77.9 | 69.2 | 78.6 | 87.5 | 91.5 | 91.4 | 62.7 | 74.2 | 72.4 |
|  | TaskRes [Yu et al., 2023] | CVPR'23 | 64.7 | 68.7 | 31.5 | 79.3 | 71.8 | 76.4 | 87.2 | 94.7 | 92.4 | 64.8 | 75.3 | 73.3 |
|  | GraphAdapter [Li et al., 2024] | NeurIPs'23 | 64.2 | 68.9 | 31.4 | 80.2 | 70.5 | 77.7 | 87.6 | 94.1 | 92.4 | 64.5 | 75.7 | 73.4 |
|  | CLIP-Adapter [Gao et al., 2024] | IJCV'24 | 62.7 | 67.5 | 26.2 | 76.7 | 67.9 | 78.0 | 87.6 | 91.7 | 91.4 | 61.0 | 73.3 | 71.4 |
|  | CLAP [Silva-Rodriguez et al., 2024] | CVPR'24 | 62.9 | 68.6 | 28.9 | 76.7 | 70.3 | 77.4 | 87.7 | 92.1 | 91.4 | 63.2 | 73.3 | 72.1 |
|  | CAA [Jiang et al., 2025] | ICCV'25 | 64.9 | 71.0 | 35.0 | 83.1 | 74.3 | 79.2 | 89.0 | 94.4 | 92.5 | 65.0 | 77.5 | 75.0 |
|  | Proposed | - | 70.8 | 74.1 | 42.3 | 81.9 | 71.4 | 86.5 | 91.6 | 96.3 | 95.5 | 68.9 | 82.6 | **78.4** |
| 16 | TIP-Adapter-F [Zhang et al., 2022] | ECCV'22 | 65.5 | 71.5 | 35.6 | 84.5 | 75.7 | 79.4 | 89.7 | 94.8 | 92.9 | 65.6 | 78.0 | 75.7 |
|  | TaskRes [Yu et al., 2023] | CVPR'23 | 63.7 | 70.7 | 36.3 | 84.0 | 76.8 | 77.6 | 87.8 | 96.0 | 93.4 | 67.1 | 78.0 | 75.8 |
|  | GraphAdapter [Li et al., 2024] | NeurIPs'23 | 65.7 | 71.2 | 36.9 | 85.3 | 76.2 | 78.6 | 88.6 | 96.2 | 93.3 | 67.6 | 78.8 | 76.2 |
|  | CLIP-Adapter [Gao et al., 2024] | IJCV'24 | 63.6 | 69.6 | 32.1 | 84.4 | 74.0 | 78.2 | 87.8 | 93.9 | 92.5 | 66.0 | 76.8 | 74.4 |
|  | CLAP [Silva-Rodriguez et al., 2024] | CVPR'24 | 65.0 | 70.8 | 33.6 | 80.1 | 75.1 | 78.5 | 88.5 | 94.2 | 91.9 | 66.4 | 76.3 | 74.4 |
|  | Ta-Adapter [Zhang et al., 2024] | PR'24 | 74.7 | 77.0 | 54.5 | 91.7 | 86.4 | 87.6 | 93.2 | 97.9 | 96.4 | 73.6 | 86.3 | **83.6** |
|  | CAA [Jiang et al., 2025] | ICCV'25 | 66.0 | 72.2 | 42.0 | 85.3 | 79.0 | 79.8 | 90.9 | 95.2 | 93.5 | 70.0 | 80.0 | 77.6 |
|  | Proposed | - | 73.1 | 75.8 | 47.2 | 87.2 | 77.0 | 86.8 | 92.5 | 97.9 | 96.1 | 73.4 | 84.4 | 81.0 |

process as for $f_{\text{train}}$. The prediction combines two complementary streams: (1) *Zero-shot stream*: the test feature is projected onto CLIP's textual prototypes, producing $\mathbf{z}_{\text{CLIP}} = \mathbf{f}_{\text{test}} W_c^\top$. (2) *Few-shot stream*: the query interacts with the learned cache $\mathcal{C} = (\Theta_{\text{train}}, L_{\text{train}})$, where keys $\Theta_{\text{train}}$ have been updated through GNN-guided refinement during training. The cosine affinity is computed as $A = \exp\big(-\beta(1 - \mathbf{f}_{\text{test}} \Theta_{\text{train}}^\top)\big)$, producing few-shot logits $\mathbf{z}_{\text{fewshot}} = A L_{\text{train}}$. The final output combines both streams in a residual formulation:

$$\mathbf{z}_{\text{final}} = \text{softmax}\big(\mathbf{z}_{\text{CLIP}} + \alpha\,\mathbf{z}_{\text{fewshot}}\big), \tag{6}$$

with $\alpha$ the same scaling factor from Eq. (4). This design preserves CLIP's zero-shot generalization while enriching predictions with relational knowledge distilled into the cache. As a result, even support examples that are not exact visual matches can guide classification through shared subspace patterns, improving few-shot adaptation without incurring additional inference cost.

## 4 EXPERIMENTAL RESULTS AND DISCUSSION

**Dataset and Experimental Setup:** We evaluate our method on 11 standard benchmarks spanning diverse visual domains: Aircraft (Maji et al., 2013), Flowers102 (Nilsback & Zisserman, 2008), SUN397 (Xiao et al., 2010), ImageNet (Deng et al., 2009), Food101 (Bossard et al., 2014), Cal-Tech101 (Fei-Fei et al., 2004), UCF101 (Soomro et al., 2012), StanfordCars (Krause et al., 2013), OxfordPets (Parkhi et al., 2012), DTD (Cimpoi et al., 2014), and EuroSAT (Helber et al., 2019). Following standard few-shot protocols, we train with 1, 2, 4, 8, and 16 labeled samples per class. Optimization is performed with AdamW, starting from a learning rate of 0.001 and scheduled by cosine annealing. We follow the protocols in (Zhang et al., 2022; Zhou et al., 2022b) and repeat the training runs independently three times for each shot configuration and report the average classification accuracy. All experiments were executed on a NVIDIA A40 GPU (48 GB).

**Implementation Details:** Our model is implemented in PyTorch. We use CLIP with ViT-B/16 (Radford et al., 2021) as the visual encoder and Transformer (Vaswani et al., 2017) as the text encoder. Following prior work (Gao et al., 2024; Zhang et al., 2022), preprocessing includes random cropping, resizing, and horizontal flipping. Our model uses two empirically tuned hyperparameters, $\alpha$ and $\beta$ (Section 3); see Appendix B for sensitivity analysis.

**Comparison with the state-of-the-art methods:** We compare our method with recent adapter-based approaches on 11 benchmarks (Table 1), including Tip-Adapter-F (Zhang et al., 2022), TaskRes (Yu et al., 2023), GraphAdapter (Li et al., 2024), CLIP-Adapter (Gao et al., 2024), and

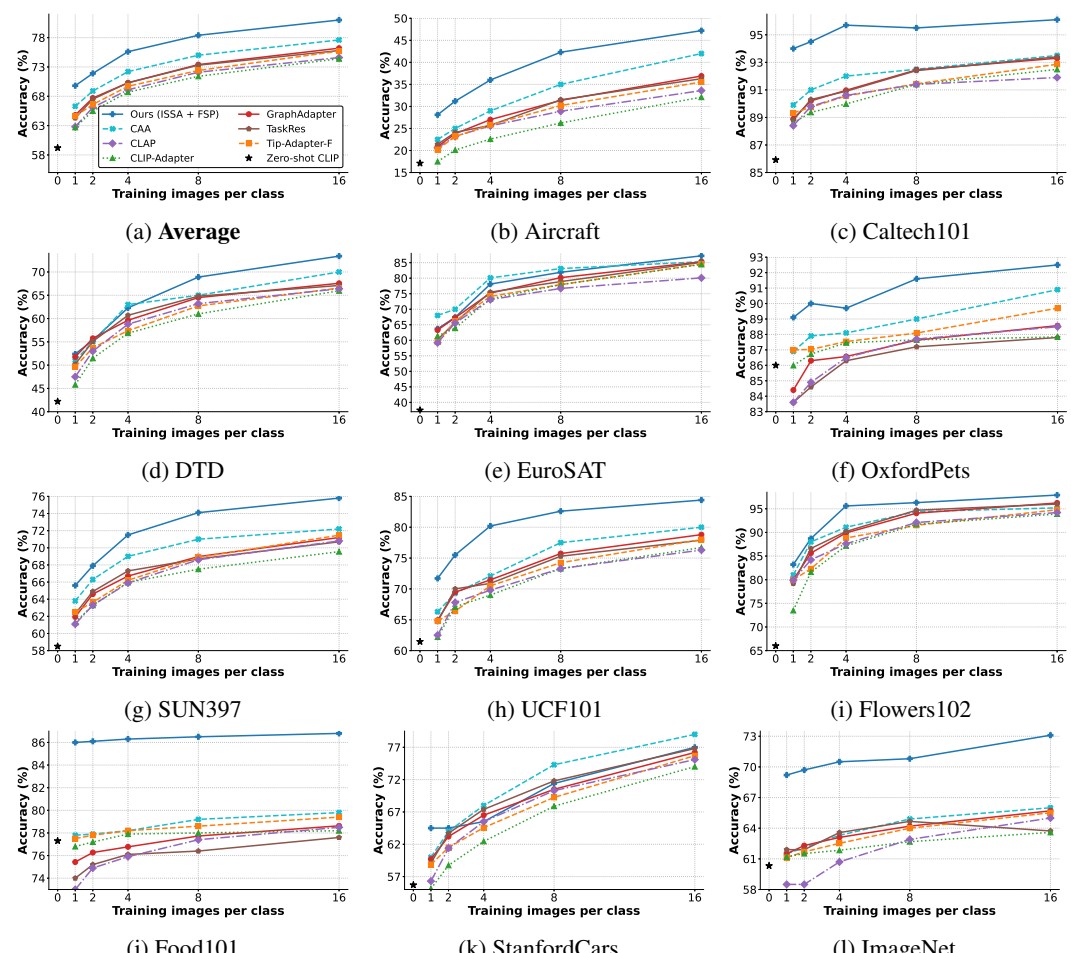

Figure 2: Accuracy (%) plots of few-shot classification on 11 benchmarks with average. We compare our graph-based feature representation against SOTA adapter-based methods (CLAP, CLIP-Adapter, GraphAdapter, TaskRes and Tip-Adapter-F) and the zero-shot CLIP baseline.

CLAP (Silva-Rodriguez et al., 2024)). Across all 1-, 2-, 4-, 8-, and 16-shot settings, our model consistently outperforms prior methods.

In the extremely low-data regime (1-shot), we obtain an average accuracy of **69.8**%, substantially outperforming the strongest competitor GraphAdapter (64.8%). Notable gains include **83.2**% on Flowers102 and **63.7**% on EuroSAT. As the number of shots increases, our model continues to demonstrate a robust generalization. With 2 shots, our model achieves **71.9**%, surpassing GraphAdapter (67.7%) and TaskRes (67.5%), with especially large improvements on Food101 (+8.0%), Aircraft (+7.0%), and UCF101 (+5.0%). In the 4-shot setting, we reach **75.6**%, outperforming GraphAdapter (70.3%), TaskRes (70.3%), and Tip-Adapter-F (69.7%), and setting new highs on Flowers102 (**95.6**, +5.4% over TaskRes), EuroSAT (**78.1**% vs. 75.2% for GraphAdapter), and ImageNet **70.5**%. Performance remains strong with 8-shot, where we achieve **78.4**%, leading on Aircraft (42.3%), Food101 (86.5%), and UCF101 (82.6%).

In the high-data regime (16-shot), our approach reaches an average of **81.0**%, surpassing GraphAdapter (76.2%) and CLAP (74.6%), and approaching Ta-Adapter (Zhang et al., 2024) (83.6%). The performance gap to Ta-Adapter is expected, as it combines prompt- and adapter-tuning scheme with additional trainable layers in both the vision and text branches of CLIP that remain active during inference. This results in stronger performance, but at the cost of added complexity and test-time overhead. Importantly, the difference is small on most datasets (Caltech **96.1%** vs. 96.4%; Pets **92.5%** vs. 93.2%; DTD **73.4%** vs. 73.6%; UCF101 **84.4%** vs. 86.3%).

In general, our method delivers state-of-the-art average performance in 1–8 shots and remains highly competitive at 16 shots (see appendix A for higher shots), while it does not require additional com-

Table 2: Accuracy (%) with varying CLIP visual encoder backbones with fixed subspaces of $|\mathcal{S}| = 8$ and the number of subspaces $|\mathcal{S}|$ with our best performing ViT-B/16 backbone for 16-shots.

| Dataset | Different Backbones for Visual Encoder | | | | Varying Number of Graph Nodes $|\mathcal{S}|$ | | | | |
|---|---|---|---|---|---|---|---|---|---|
| | RN50 | RN101 | ViT-B/32 | ViT-B/16 | 4 | 8 | 16 | 32 | 64 |
| Aircraft | 39.7 | 41.7 | 40.4 | **47.2** | 46.5 | **47.2** | 46.7 | 46.9 | 45.8 |
| Flowers | 95.6 | 96.3 | 95.9 | **97.9** | 97.6 | **97.9** | 97.8 | 97.8 | 97.8 |
| UCF101 | 78.5 | 81.2 | 81.9 | **84.4** | 84.4 | **84.4** | 84.3 | 84.2 | 84.2 |
| EuroSAT | 78.1 | 79.0 | 76.5 | **87.2** | 83.6 | **87.2** | 84.2 | 76.3 | 76.2 |
| DTD | 65.8 | 67.3 | 65.8 | **73.4** | 69.6 | **73.4** | 70.0 | 69.0 | 68.8 |

putation during testing. The gains generalize across diverse domains, including fine-grained (Aircraft, Pets, Cars, Food, Flowers102), satellite (EuroSAT), texture (DTD), and general-object (Caltech101, ImageNet) benchmarks. Figure 2 further illustrates that our graph-refined model consistently achieves higher accuracy as shots increase, confirming the effectiveness of cache refinement without inference overhead.

**Ablation Study:** We conduct ablation study on Aircraft, Flowers102, UCF101, DTD and EuroSAT to evaluate the contribution of each component of our model. Specifically, we analyze the effect of the visual backbone, subspace granularity, intra-sample modeling (ISSA), feature subspace propagation (FSP), and key graph hyperparameters. Results are summarized in Tables 2-7.

*Effect of Visual Backbone Architecture:* Table 2 (left) reports results with different CLIP visual encoders: ResNet-50 (He et al., 2016), ResNet-101 (He et al., 2016), ViT-B/32 (Dosovitskiy et al., 2021), and ViT-B/16 (Dosovitskiy et al., 2021). Accuracy improves consistently with backbone strength: on Aircraft, from 39.7% (ResNet-50) to 41.7% (ResNet-101) and 47.2% (ViT-B/16); on UCF101 from 78.5% to 84.4% and on Flowers102 from 95.6% to 97.9%. These gains indicate that stronger backbones provide more discriminative features, which our graph refinement further amplifies. This explains why most SOTA methods in Table 1 adopt transformer-based CLIP backbones. More importantly, the consistent improvement across all architectures highlights the *generality* of our approach. Regardless of the backbone, our graph-driven refinement yields superior performance.

*Effect of Graph Node (subspace) Granularity:* We analyze the impact of subspace granularity by varying the number of graph nodes $|\mathcal{S}| \in \{4, 8, 16, 32, 64\}$ with ViT-B/16 as the backbone (Table 2, right). An intermediate value of $|\mathcal{S}| = 8$ provides the best balance between feature detail and complexity, achieving the highest accuracy in all datasets. On Aircraft, accuracy increases from 46. 5% ($|\mathcal{S}| = 4$) to 47.2% ($|\mathcal{S}| = 8$), but degrades as $|\mathcal{S}|$ increases further. Similar trends appear, though less pronounced, for Flowers102 and UCF101. In particular, fine-grained domains like Aircraft are more sensitive to subspace resolution, while broader domains (UCF101) remain relatively robust. This suggests that appropriately chosen subspace granularity is critical for maximizing performance.

Table 3: Impact of varying model hyperparmeters on Accuracy (%) for 16-shots. From left to right: Teleport Probability $\gamma$, Propagation Steps T, and Aggregation modes. Best are shown in **Bold**.

| Dataset | Teleport Probability $\gamma$ | | | | | Propagation Steps $T$ | | | | | Aggregation Mode | | | | | |
|---|---|---|---|---|---|---|---|---|---|---|---|---|---|---|---|---|
| | 0.1 | 0.3 | 0.5 | 0.7 | 0.9 | 1 | 3 | 5 | 7 | 9 | attn | cat | proj | mean | max | sum |
| Aircraft | **47.2** | 47.0 | 46.9 | 46.8 | 46.8 | **47.2** | 46.9 | 46.3 | 46.3 | 46.1 | **47.2** | 46.6 | 46.7 | 46.5 | 46.7 | 46.6 |
| Flowers | **97.9** | 97.5 | 97.5 | 97.1 | 97.0 | **97.9** | 97.7 | 97.2 | 96.4 | 96.8 | **97.9** | 97.3 | 97.2 | 97.4 | 97.6 | 97.7 |
| UCF | **84.4** | 84.3 | 84.1 | 84.2 | 84.1 | **84.4** | 84.2 | 84.0 | 84.0 | 83.8 | **84.4** | 84.1 | 84.0 | 83.8 | 83.5 | 83.8 |
| EuroSAT | **87.2** | 87.0 | 85.9 | 85.4 | 84.9 | **87.2** | 85.1 | 83.9 | 82.8 | 82.2 | **87.2** | **87.2** | 87.0 | 86.3 | 85.7 | 85.5 |
| DTD | **73.4** | 70.3 | 70.0 | 70.0 | 69.5 | **73.4** | 73.0 | 73.0 | 72.6 | 71.9 | **73.4** | 73.1 | 72.8 | 70.0 | 69.7 | 69.5 |

*Impact of Graph Hyperparameters:* We further analyze key hyperparameters on Aircraft (16-shot, Table 3). A low teleport probability ($\gamma = 0.1$) achieves the best accuracy (47.2%), highlighting the importance of the *propagation term* for modeling subspace relationships. As Eq. (3) shows, smaller $\gamma$ increases the weight $(1 - \gamma)$ of the *propagation term*, justifying the importance of relational modeling. For propagation step $T$, a single iteration is optimal: deeper propagation ($T > 3$) leads to oversmoothing and accuracy degradation. For ISSA aggregation in Eq. (1), attention achieves the highest accuracy (47.2%), outperforming concatenation, linear projection, and various pooling (max, mean, and sum). Overall, higher importance to *propagation term* with adaptive attention maximizes the quality of the representation while avoiding over-smoothing. For ISSA aggregation in Eq. (1), attention achieves the highest accuracy outperforming both "cat" and "proj" across datasets. On Aircraft, **attn** (47.2%) exceeds **cat** (46.6%) and **proj** (46.7%), and the same trend appears on DTD (**attn**: 73.4%, **cat**: 73.1%, **proj**: 72.8%). Unlike "cat" or "proj", which apply a fixed, data-

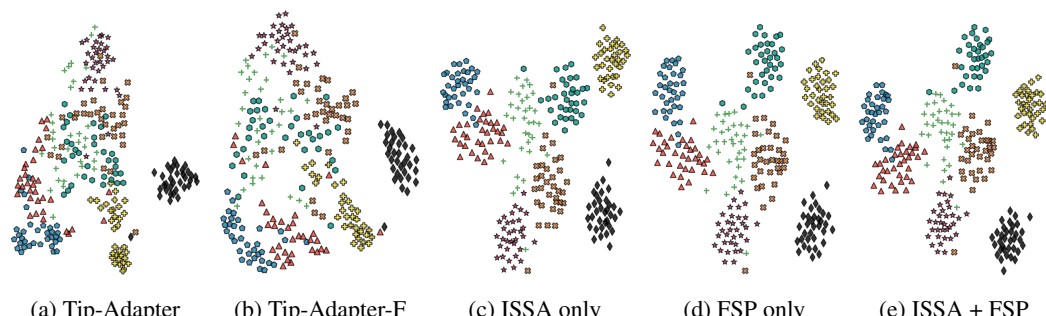

|  (a) Tip-Adapter | (b) Tip-Adapter-F | (c) ISSA only | (d) FSP only | (e) ISSA + FSP |

Figure 3: t-SNE (Van Der Maaten, 2014) visualizations of different approaches on the Aircraft dataset, illustrating class separability and compactness of logits (see Eq. (4)), computed over 8 randomly selected classes. (a) Tip-Adapter (Zhang et al., 2022), (b) Tip-Adapter-F (Zhang et al., 2022), (c) ISSA only, (d) FSP only, (e) ISSA + FSP.

independent mixing of neighborhood subspace statistics (mean/max/std), our attention mechanism learns *content-dependent* weights that adaptively reweight and combine these complementary cues. This adaptive fusion yields more stable and expressive subspaces without oversmoothing.

*Effect of ISSA and FSP:* We evaluate the contribution of the **Inductive Statistical Subspace Aggregation** (ISSA) and **Feature Subspace Propagation** (FSP) modules in our graph-driven representation learning for knowledge incorporation framework (Table 4). At 16 shots, ISSA consistently improves performance: on Aircraft, it alone achieves 46.9%, outperforming FSP-only (46.4%), and reaches 47.2% when combined. On UCF101, ISSA (83.3%) is comparable to FSP (at 83.6%) but increases to 84. 2% with both. On Flowers102, ISSA trails FSP (97.3% vs. 97.6%) yet achieves the best result when combined (97.9%). These findings highlight ISSA's role in capturing *intra-sample dependencies*, while FSP contributes complementary *global context*. Removing both modules reduces the model to Tip-Adapter-F (no graph-driven refinement), causing sharp drops at 16-shot: Aircraft falls from 47.2% to 35.6%, UCF101 from 84.2% to 78.0%, and Flowers102 from 97.9% to 94.8%. It is worth mentioning that under extremely limited data (e.g., 1-shot on UCF101), FSP-only slightly outperforms the combined model (72.0% vs 71.7%), suggesting that global propagation may introduce noise when support evidence is minimal.

Overall, combining ISSA and FSP consistently yields the highest accuracy across datasets, confirming their complementary roles in improving feature representation for cache refinement. This is further supported by the t-SNE visualizations in Figure 3, which show more compact and well-separated class clusters when both modules are active. More plots are provided in the Appendix D.

Table 4: Accuracy (%) of our key GNN modules with varying $K$-shots on five datasets.

| Dataset | ISSA | FSP | $K = 16$ | $K = 8$ | $K = 4$ | $K = 2$ | $K = 1$ |
|---------|------|-----|----------|---------|---------|---------|---------|
| Aircraft | × | × | 35.6 | 30.2 | 25.8 | 23.2 | 20.2 |
|  | × | ✓ | 46.4 | 41.6 | 35.5 | 30.6 | 28.1 |
|  | ✓ | × | 46.9 | 42.1 | 35.2 | 30.8 | **28.2** |
|  | ✓ | ✓ | **47.2** | **42.3** | **36.0** | **31.2** | 28.1 |
| Flowers | × | × | 94.8 | 91.5 | 88.8 | 82.3 | 80.0 |
|  | × | ✓ | 97.6 | 96.2 | 95.2 | 87.8 | 81.9 |
|  | ✓ | × | 97.3 | 96.1 | 95.4 | 88.1 | 82.6 |
|  | ✓ | ✓ | **97.9** | **96.3** | **95.6** | **88.7** | **83.2** |
| UCF101 | × | × | 78.0 | 74.2 | 70.6 | 66.4 | 64.9 |
|  | × | ✓ | 83.6 | 82.1 | 79.9 | 75.2 | **72.0** |
|  | ✓ | × | 83.3 | 82.2 | 79.2 | 75.4 | 71.1 |
|  | ✓ | ✓ | **84.2** | **82.6** | **80.2** | **75.5** | 71.7 |
| EuroSAT | × | × | 84.5 | 77.9 | 74.1 | 66.1 | 59.5 |
|  | × | ✓ | 79.1 | 78.5 | 76.3 | 66.2 | 60.5 |
|  | ✓ | × | 81.9 | 76.0 | 74.7 | 66.9 | 60.7 |
|  | ✓ | ✓ | **87.2** | **81.9** | **78.1** | **67.4** | **63.7** |
| DTD | × | × | 66.5 | 62.7 | 57.4 | 53.7 | 49.6 |
|  | × | ✓ | 71.2 | 66.1 | 59.5 | 55.0 | 50.9 |
|  | ✓ | × | 70.9 | 66.3 | 59.1 | 53.5 | 51.1 |
|  | ✓ | ✓ | **73.4** | **68.9** | **62.3** | **55.3** | **52.4** |

Table 5: Impact of modeling relations between subspaces: with (w) vs. without (w/o) edges.

| Shot | Edges | Aircraft | Flowers102 | UCF101 | EuroSAT | DTD |
|------|-------|----------|------------|--------|---------|-----|
| 1-shot | w/o | 26.7 | 76.41 | 70.65 | 54.7 | 50.1 |
|  | w | **28.1** | **83.2** | **71.7** | **63.7** | **52.4** |
| 4-shot | w/o | 30.5 | 84.5 | 75.7 | 61.4 | 51.6 |
|  | w | **36.0** | **95.6** | **80.2** | **78.1** | **62.3** |
| 16-shot | w/o | 37.9 | 83.47 | 73.4 | 68.7 | 58.7 |
|  | w | **47.2** | **97.9** | **84.4** | **87.2** | **73.4** |

Table 6: Inter-slice similarity before/after refinement (left) and accuracy under contiguous vs. random partitioning (right). Full table in Appendix C.

| Dataset | # Splits | Inter-slice Sim. (16-shot) | | # Shot | Partitioning (8-splits) | |
|---------|----------|---------|-------|--------|-----------|--------|
|  |  | Before | After |  | Contiguous | Random |
| Aircraft | 4 | −0.0001 | **0.9963** | 4 | **36.0** | 35.5 |
|  | 16 | −0.0066 | **0.9982** | 16 | **47.2** | 47.0 |
| Flowers102 | 4 | 0.0214 | **0.9965** | 4 | **95.6** | 95.5 |
|  | 16 | 0.0194 | **0.9977** | 16 | **97.9** | 97.8 |

*Impact of Modeling Relations Between Subspaces:* Table 5 shows that explicitly modeling inter-subspace relations consistently outperforms the edge-free variant (MLP) across datasets and shot settings. On Flowers102 at 1-shot, accuracy improves from 76.41% to 83.2%; on Aircraft, from

26.7% to 28.1%; and on UCF101, from 70.65% to 71.7%. The gap widens with more shots, for example, on Aircraft, our method achieves 47.2% versus 37.9% (+9.3%). These results confirm that subspace graph refinement, which enables subspace interactions, yields consistent improvements, especially for fine-grained categories and in lower-shot regimes.

*Effect of Graph Refinement on Inter-slice Similarity:* Table 6 reports the effect of graph refinement (ISSA+FSP) by measuring the average cosine similarity between partitioned subspaces of CLIP features in the 16-shot setting. Before refinement, slices show low or negative similarity (e.g., Aircraft: $-0.0001$, Flowers102: $0.0214$), indicating that they disagree in direction. After refinement, similarity rises sharply to $\approx 0.99$ (Aircraft: $0.9963$, Flowers102: $0.9985$), meaning that subspaces agree on the same image-level semantics, not that features across different images have collapsed. High cosine indicates that slices lie in a tight semantic cone; their norms and higher-order structures still differ, and the refined slices are concatenated to form $\tilde{f}_{\text{train}}$ with preserved fine variation. This reflects a shift from anisotropy to more coherent semantics. In few-shot regimes, such mutually reinforcing subspaces stabilize predictions and regularize CLIP's embedding toward a task-aligned representation space. CLIP's vision–language alignment remains intact since we never fine-tune its encoders; the graph only distills relational cues into the cache keys.

*Effect of Partitioning Strategy:* We compare contiguous versus random partitioning of the feature vector into subspaces (Table 6). Both strategies yield very similar performance (e.g., 47.2% vs. 47.0% on Aircraft, 97.9% vs. 97.8% on Flowers102 in 16-shot), indicating that the graph can learn useful relationships regardless of how dimensions are grouped, as long as all are covered. We keep contiguous chunks as the default because they are simple to implement and guarantee non-overlapping coverage of the feature space, but we emphasize that the small gaps between contiguous and random splits reflect robustness, not a strong prior that CLIP's channel ordering is semantically meaningful. See full table in Appendix C

**Computational efficiency:** We benchmark the overhead of our graph-driven refinement (ISSA + FSP) against Tip-Adapter-F (Zhang et al., 2022) in Table 7. Tip-Adapter-F has $\sim 0.82$M parameters and requires $1.64 \times 10^{-3}$ GFLOPs per sample, while our graph adds only $\sim 0.037$M parameters and $1.12 \times 10^{-3}$ GFLOPs. This yields a total of $\sim 0.86$M and $2.75 \times 10^{-3}$ GFLOPs on ImageNet with 8 subspaces. In a 16-shot setting with 512-d CLIP features and batch size 256, Tip-Adapter-F uses 2932 MB GPU memory during training, compared to 2940 MB for our model, a negligible $\sim 0.3\%$ overhead on an NVIDIA A40. Training time per epoch increases modestly (13s vs. 6s), which remains practical given the accuracy gains. More importantly, at inference we discard the GNN and rely solely on the refined cache, matching the test-time efficiency of Tip-Adapter-F.

Table 7: Efficiency comparison: Ours vs. Tip-Adapter-F Zhang et al. (2022) on ImageNet 16-shot.

|  | Acc. (%) | Param. (M) | G-Flops | Peak Memory (MB) | Train-time/epoch (sec) |
|---|---|---|---|---|---|
| Tip-Adapter-F | 65.5 | 0.82 | $1.64 \times 10^{-3}$ | 2932 | 6 |
| Ours | 73.1 | 0.86 | $2.75 \times 10^{-3}$ | 2940 | 13 |

## 5    CONCLUSION

We introduced a graph-driven framework for few-shot adaptation of CLIP, addressing the limitations of frozen embeddings in capturing fine-grained variation under scarce supervision. Our design partitions CLIP features into statistical subspaces and refines them through Inductive Statistical Subspace Aggregation (ISSA) and Feature Subspace Propagation (FSP), resulting in enriched cache keys that encode both local and global contextual cues. Importantly, this refinement operates only during training, so inference remains graph-free, lightweight, and as efficient as existing cache-based methods. Extensive experiments across 11 benchmarks demonstrate consistent state-of-the-art performance in the 1–8 shot regime and competitive results at 16 shots, with negligible computational overhead. Ablations further validate the complementary roles of ISSA and FSP, the importance of subspace granularity, and the robustness of our method across backbones and domains.

By showing how intra-feature graph refinement can be distilled into cache representations, our work bridges representation learning and semi-parametric adaptation, opening new directions for exploiting structure in frozen vision–language models. Although the method scales well on standard few-shot benchmarks, it remains to be tested on larger support sets or higher-resolution inputs, where graph-based training steps may pose memory or runtime challenges. Future extensions could explore task-aware subspace partitioning, integration with other large-scale multimodal models, and broader applications beyond classification.

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

# APPENDIX

This appendix presents extended results and analyses supporting the main paper, including evaluations for 32- and 64-shot settings, hyperparameter studies, and additional t-SNE visualizations. We also disclose the use of LLMs as an assisting tool.

## A  EXTENDED EVALUATION UNDER HIGH-SHOT SETTINGS

Table 8: Few-shot performance (%) across varying shots on five datasets, including extended results for 32- and 64-shot settings.

| Dataset | 1-shot | 2-shot | 4-shot | 8-shot | 16-shot | 32-shot | 64-shot |
|---|---|---|---|---|---|---|---|
| Flowers102 | 83.2 | 88.7 | 95.6 | 96.3 | 97.9 | 98.2 | 98.5 |
| UCF101 | 71.7 | 75.5 | 80.2 | 82.6 | 84.4 | 86.1 | 87.3 |
| Aircraft | 28.1 | 31.1 | 36.0 | 42.3 | 47.2 | 54.8 | 52.3 |
| EuroSAT | 63.7 | 67.4 | 78.1 | 81.9 | 87.2 | 87.6 | 89.4 |
| DTD | 52.4 | 55.3 | 62.3 | 68.9 | 73.4 | 73.2 | 75.8 |

To further assess the scalability and robustness of our graph-refined cache model, we extend our evaluation beyond the standard 1–16 shot regime to include 32-shot and 64-shot settings. Table 8 presents the classification performance across seven shot levels on five representative datasets: Aircraft, Flowers102, UCF101, EuroSAT and DTD. We observe that our method continues to exhibit strong performance trends with increasing data availability. On Flowers102, accuracy rises from 83.2% at 1-shot to 98.5% at 64-shot, demonstrating near-saturation and confirming the model's ability to fully exploit additional supervision in fine-grained domains. UCF101, a dynamic and diverse action recognition dataset, shows a steady gain from 71.7% at 1-shot to 87.3% at 64-shot, reflecting effective generalization over a wide range of motion patterns and classes. Interestingly, while Aircraft improves significantly up to 32-shot (from 28.1% at 1-shot to 54.8%), we observe a slight dip at 64-shot (52.3%). This plateau suggests that very fine-grained tasks may benefit more from moderate support sizes where overfitting is minimized and inter-class similarity does not overshadow cache discriminability. We hypothesize that beyond a certain point, adding more examples may introduce noise or redundancy in the support set for datasets with high intra-class similarity and subtle inter-class variation. Overall, these results affirm that our method scales reliably with increasing supervision, while maintaining high retrieval quality across low- and high-shot regimes. The performance stability beyond 16-shot further supports the general applicability of our model to both constrained and data-rich few-shot scenarios.

## B  HYPERPARAMETER ANALYSIS

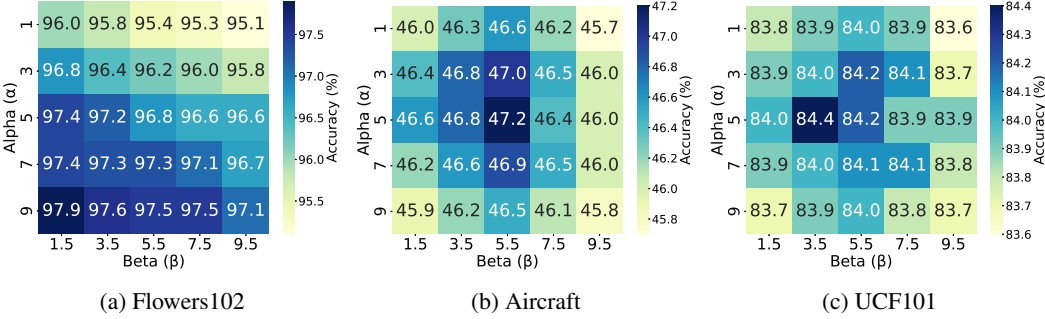

(a) Flowers102      (b) Aircraft      (c) UCF101

Figure 4: Accuracy heatmaps from grid search over $\alpha$ and $\beta$ on (a) Flowers102, (b) Aircraft, and (c) UCF101, illustrating the impact of residual weighting and affinity sharpness on cache performance.

We analyze the effect of cache hyperparameters ($\alpha$, $\beta$) on performance. We visualize the effect of the two key cache hyperparameters, $\alpha$ and $\beta$, using heatmaps in Figure 4, which show classification

accuracy across grid search on three representative datasets: Flowers102, Aircraft, and UCF101. These datasets cover fine-grained, structural, and dynamic domains, respectively. On Flowers102, a visually discriminative and fine-grained dataset, accuracy improves steadily with higher values of both $\alpha$ and $\beta$, reaching a peak of 97.9% at $\alpha = 9$ and $\beta = 1.5$. For Aircraft, which demands fine structural detail recognition, the optimal accuracy of 47.2% is observed at $\alpha = 5$ and $\beta = 5.5$, reflecting a more balanced reliance between CLIP priors and support features. UCF101, which includes motion-rich action classes, benefits from sharper affinity ($\beta = 3.5$) and a moderately weighted cache ($\alpha = 5$), achieving a peak of 84.4%. These observed trends motivate our approach to hyperparameter tuning. In our GNN-based cache framework, $\alpha$ controls the residual weighting between zero-shot CLIP logits and cache-based retrieval, while $\beta$ determines the sharpness of similarity weighting during affinity computation. To make this interaction both expressive and generalizable, we follow a two-stage procedure. During training, we use fixed initial values for the hyperparameters: $\alpha$ is set to 10 for Flowers102, 5 for Aircraft, and 3 for UCF101, while $\beta$ is uniformly set to 1 across all datasets. After training, a grid search is conducted on the validation set to identify the optimal $(\alpha, \beta)$ pair, which is then used for final evaluation on the test set. This post-training tuning process introduces no additional inference-time overhead and enables the model to flexibly adapt to the characteristics of each dataset. Overall, the cache model's robustness hinges on these interpretable and tunable parameters. Their complementary influence enables our framework to flexibly balance prior knowledge and task specificity, consistently yielding high accuracy across fine-grained, structured, and dynamic classification scenarios.

## C  ADDITIONAL RESULTS RELATED TO INTER-SLICE COSINE SIMILARITY AND RANDOM PARTITION

*Effect of Graph Refinement on Inter-slice Cosine Similarity:* We assess the effect of our graph-based refinement (ISSA + FSP) by computing average pairwise cosine similarity between partitioned CLIP subspaces before and after refinement. Table 9 shows pre-refinement similarities are weak or negative (e.g., Aircraft: $-0.0066$ at 2 splits, $-0.0021$ at 8 splits; Flowers102: $-0.0074$ at 2 splits, $-0.0041$ at 8 splits), indicating poor alignment. Post-refinement values rise sharply toward 0.99

Table 9: Average inter-slice cosine similarity before and after graph refinement

| Dataset | # Splits | Before | After |
|---------|----------|--------|-------|
| Aircraft | 2 splits | $-0.0066$ | **0.9960** |
| | 4 splits | $-0.0001$ | **0.9963** |
| | 8 splits | $-0.0021$ | **0.9969** |
| | 16 splits | $-0.0066$ | **0.9982** |
| Flowers102 | 2 splits | $-0.0074$ | **0.9810** |
| | 4 splits | $0.0214$ | **0.9965** |
| | 8 splits | $-0.0041$ | **0.9972** |
| | 16 splits | $0.0194$ | **0.9977** |
| UCF101 | 2 splits | $-0.0037$ | **0.9801** |
| | 4 splits | $0.0294$ | **0.9744** |
| | 8 splits | $0.0053$ | **0.9769** |
| | 16 splits | $0.0227$ | **0.9550** |
| EuroSAT | 2 splits | $-0.0007$ | **0.9966** |
| | 4 splits | $0.0091$ | **0.9893** |
| | 8 splits | $0.0054$ | **0.9965** |
| | 16 splits | $-0.0195$ | **0.9965** |
| DTD | 2 splits | $-0.0128$ | **0.9968** |
| | 4 splits | $-0.0092$ | **0.9678** |
| | 8 splits | $-0.0090$ | **0.9721** |
| | 16 splits | $-0.0370$ | **0.9166** |

Table 10: Comparison of few-shot classification accuracy (%) using contiguous vs. random partitioning into 8 subspaces. Results are shown for 1-, 4-, and 16-shot settings on five datasets. Random chunking performs comparably, indicating robustness to partition strategy.

| Setting | Contiguous Chunk | Random Chunk |
|---|---|---|
| Aircraft (1-shot) | **28.1** | 28.0 |
| Aircraft (4-shot) | **36.0** | 35.5 |
| Aircraft (16-shot) | **47.2** | 47.0 |
| Flowers102 (1-shot) | 83.2 | **83.3** |
| Flowers102 (4-shot) | **95.6** | 95.5 |
| Flowers102 (16-shot) | **97.9** | 97.8 |
| UCF101 (1-shot) | **71.7** | 70.7 |
| UCF101 (4-shot) | **80.2** | 79.5 |
| UCF101 (16-shot) | **84.4** | 84.0 |
| EuroSAT (1-shot) | **63.7** | 63.0 |
| EuroSAT (4-shot) | **78.1** | 77.5 |
| EuroSAT (16-shot) | **87.2** | 87.0 |
| DTD (1-shot) | **52.4** | 52.0 |
| DTD (4-shot) | **62.3** | 58.6 |
| DTD (16-shot) | **73.4** | 70.8 |

across all splits and datasets (e.g., Aircraft: 0.9960 at 2 splits to 0.9982 at 16 splits; Flowers102: 0.9810 at 2 splits to 0.9977 at 16 splits), reflecting strong semantic coherence.

This trend is consistent across both datasets and improves with finer subspace granularity, for instance, on Aircraft, similarity increases from 0.9960 (2 splits) to 0.9982 (16 splits), and on Flowers102 from 0.9810 to 0.9977. These results confirm that our graph-driven refinement effectively models intra-feature context, enhancing few-shot discriminability.

*Effect of Partitioning Strategy:* We compare contiguous and random partitioning for dividing the feature vector into 8 subspaces. Contiguous chunks and non-overlapping, covering the entire vector, while random chunks are sampled from arbitrary start positions and may overlap or leave gaps.

Table 10 show contiguous partitioning provides a consistent gain (e.g., Aircraft 1-shot: 28.1% vs. 28.0%, 4-shot: 36.0% vs. 35.5%, Flowers102 16-shot: 97.9% vs. 97.8%) over random partitioning strategy. The performance drop with random chunks may result from uneven coverage, potentially missing important dimensions needed for few-shot discriminability.

## D ADDITIONAL VISUALIZATIONS

This appendix further provides additional visualizations to qualitatively support our findings. Figure 5-9 presents t-SNE visualizations on five datasets: Flowers102, UCF101, CalTech101, DTD, and EuroSAT, to qualitatively assess the impact of ISSA and FSP. We compare the feature distributions of our model variants against baselines such as Tip-Adapter (Zhang et al., 2022) and Tip-Adapter-F (Zhang et al., 2022). Across most datasets, Tip-Adapter and Tip-Adapter-F exhibit loosely grouped clusters with noticeable class overlap. Applying ISSA or FSP individually leads to more compact and structured representations. Their combination yields the clearest class separation, reflecting the complementary effects of ISSA and FSP.

Interestingly, on CalTech101 (see Figure 9), both Tip-Adapter and Tip-Adapter-F already produce well-separated clusters, likely due to the dataset's visually distinctive object categories. This observation aligns with the high accuracy achieved by all methods in Table 1. However, our method further enhances cluster compactness and class separability, offering a consistent improvement even over these already strong-performing approaches. These visualizations confirm that ISSA and FSP not only benefit challenging datasets but also strengthen representations when initial clustering is already well-structured.

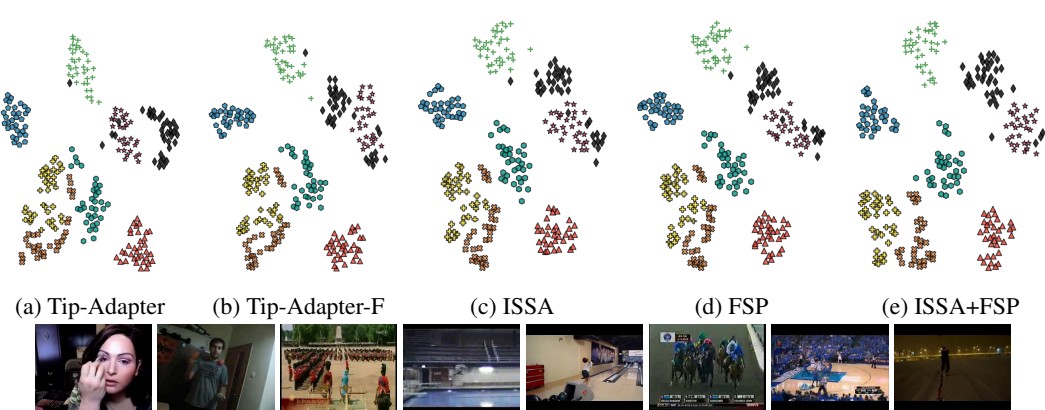

(a) Tip-Adapter     (b) Tip-Adapter-F     (c) ISSA     (d) FSP     (e) ISSA+FSP

Figure 5: t-SNE visualizations of feature distributions for 8 randomly selected classes from the UCF101 dataset using (a) Tip-Adapter (Zhang et al., 2022), (b) Tip-Adapter-F (Zhang et al., 2022), (c) ISSA, (d), FSP (e) and Ours combined ISSA+FSP method (e). Compared to the Tip-Adapter variants, ISSA and FSP individually result in more compact clusters, while their combination achieves the most distinct class separation. The bottom row displays representative video frames from each of the 8 selected classes, illustrating the visual diversity and complexity present in the dataset.

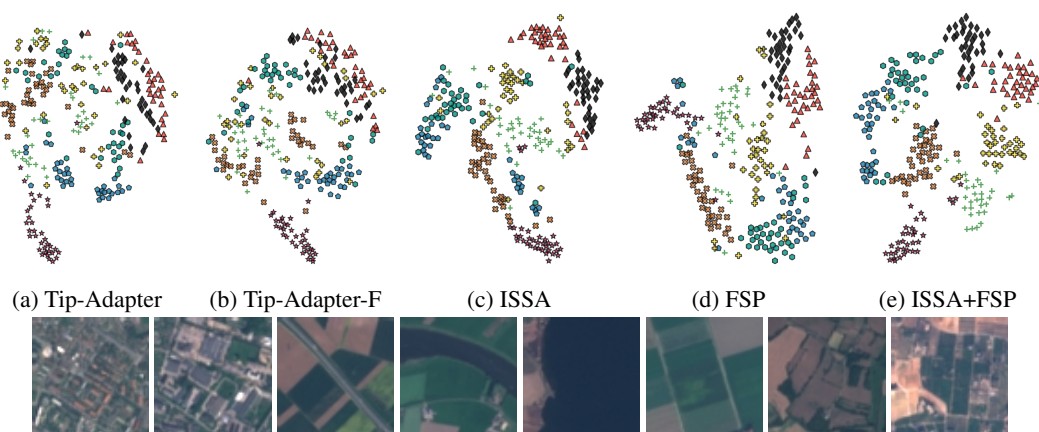

(a) Tip-Adapter     (b) Tip-Adapter-F     (c) ISSA     (d) FSP     (e) ISSA+FSP

Figure 6: t-SNE visualizations of feature distributions for 8 randomly selected classes from the EuroSAT dataset using (a) Tip-Adapter (Zhang et al., 2022), (b) Tip-Adapter-F (Zhang et al., 2022), (c) ISSA, (d) FSP, and (e) our combined ISSA+FSP method. The combined approach yields more distinct and compact clusters, indicating improved spatial feature discrimination. The bottom row displays representative satellite images from the selected classes, illustrating the diversity in terrain and land cover types.

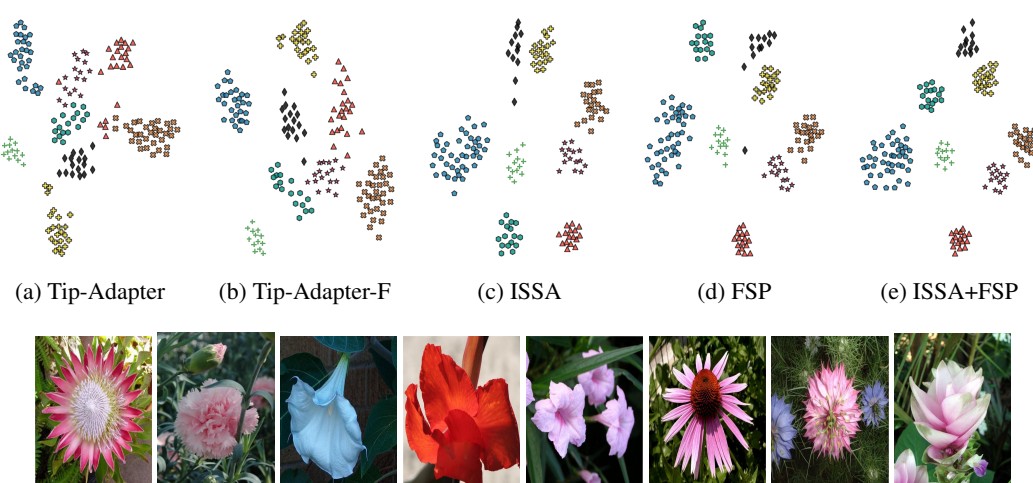

Figure 7: t-SNE visualizations of feature distributions for 8 randomly selected classes from the Flowers102 dataset using (a) Tip-Adapter (Zhang et al., 2022), (b) Tip-Adapter-F (Zhang et al., 2022), (c) ISSA, (d) FSP, and (e) our combined ISSA+FSP method. The combined approach produces the most compact and well-separated clusters, reflecting enhanced class discriminability in this fine-grained setting. The bottom row shows representative flower images from the selected classes, illustrating subtle variations in color, shape, and structure that challenge few-shot recognition.

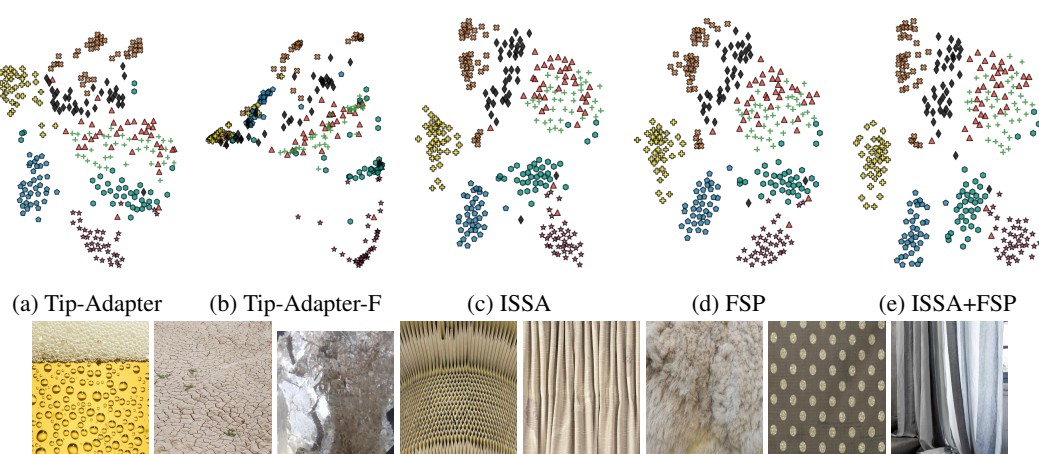

Figure 8: t-SNE visualizations of feature distributions for 8 randomly selected classes from the DTD dataset using (a) Tip-Adapter (Zhang et al., 2022), (b) Tip-Adapter-F (Zhang et al., 2022), (c) ISSA, (d) FSP, and (e) our combined ISSA+FSP method. The combined model produces the most distinct and compact clusters, indicating improved discrimination of fine-grained texture patterns. The bottom row displays representative texture images from the selected classes, capturing the diversity in structural and material properties across the dataset.

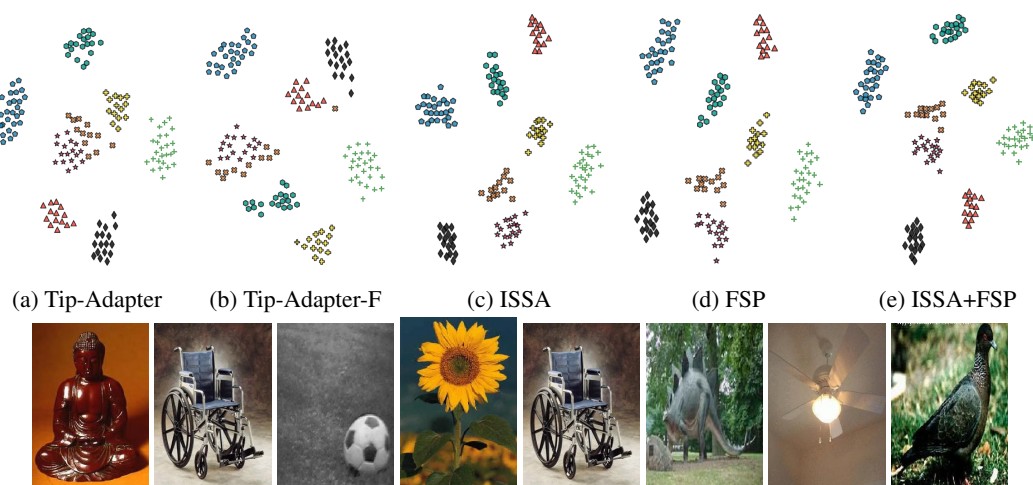

(a) Tip-Adapter    (b) Tip-Adapter-F    (c) ISSA    (d) FSP    (e) ISSA+FSP

Figure 9: t-SNE visualizations of feature distributions for 8 randomly selected classes from the CalTech101 dataset using (a) Tip-Adapter (Zhang et al., 2022), (b) Tip-Adapter-F (Zhang et al., 2022), (c) ISSA, (d) FSP, and (e) our combined ISSA+FSP method. Even with Tip-Adapter variants showing relatively clean separation, our combined approach achieves the most compact and distinct clusters. The bottom row shows representative images from each class, reflecting the visual diversity and object-centric nature of the dataset.

# E    LLM USAGE

We used a large language model (LLM) as a general-purpose assist tool only for light copyediting (fixing grammar, spelling, and phrasing) to improve readability. It did not help with research ideas, methods, experiments, analysis, or substantive writing and should not be considered a contributor or author. All text and results were written and checked by the authors, and any edits suggested by the LLM were accepted only after human review.

