# OpenReview forum: "Graph-Refined Representation Learning for Few-Shot Classification via CLIP Adaptation"
_ICLR.cc/2026/Conference — ICLR 2026 Conference Desk Rejected Submission_

### Official Review · Reviewer_JNSy · 2025-11-01

**Soundness:** 3
**Presentation:** 3
**Contribution:** 3
**Rating:** 6
**Confidence:** 4

**Summary:**

This paper presents a novel graph-driven cache refinement framework that improves CLIP’s prior knowledge for lightweight few-shot adaptation onto downstream tasks. The proposed method designs a two-stage graph learning approach. For each feature sub-vector (named subspace in this paper), in the first stage, the Inductive Statistical Subspace Aggregation (ISSA) updates it by summarizing neighborhood statistical information. Then, in the second stage, each subspace is further enhanced by absorbing contextual signals from other subspaces through the Feature Subspace Propagation (FSP) operation.

**Strengths:**

1. This paper is well organized and easy to follow. In addition, the motivation of this work is well clarified.

2. The idea of dividing the whole feature vector into different subspaces and constructing the graph between those subspaces is interesting.

3. The authors have conducted extensive experiments, and the experimental results indicate the effectiveness of the proposed method.

**Weaknesses:**

1. One of my major concerns lies in the training process of the proposed method. I cannot understand how the graph-driven knowledge influences the optimization of the cache keys. Specifically, the authors claim that the keys $\Theta_{train}$ “are initialized with the L2-normalized CLIP visual features from the support set, and updated via gradient during training”, but it can be observed from Eq. (4) that the graph-enhanced features $\tilde f_{train}$ do not influence the training of $\Theta_{train}$, which is only optimized by the first term of Eq. (4). In addition, the second term in Eq. (4) only trains the parameters of the constructed graph, while those parameters of CLIP are frozen, but this graph is not utilized in the inference stage, so I do not know how does this term affect the cache keys and final performance.

2. In the Inductive Statistical Subspace Aggregation operation, the authors claim that the aggregation of neighborhood statistical information can “capture fine-grained relational dependencies and discover robust feature representations”. I am wondering why the neighborhood statistical information could help to extract robust feature representations.

3. For the experimental results, the competing benchmarks appear to be outdated. Could the authors compare their proposed method with those published in 2025? Such as in CVPR 2025, ICML 2025, ICCV 2025, or ICLR 2025?

4. I am confused about the results in Table 6. If I am misunderstanding, please correct me. From Table 6, we can find that the cosine similarities between different partitioned subspaces are nearly equal to 1. Does this mean that those subspaces are becoming more redundant since they become more similar? If so, it may indicate that the graph refinement operation can lead to information loss, since different subspaces (i.e., feature dimensions) cannot represent distinct visual information. Furthermore, from the results, we can see that the graph refinement will modify the space of visual features, it may degrade the vision-language alignment of CLIP model, so will the graph refinement influence the effectiveness of the pre-trained CLIP text-based classifier? Could the authors provide more explanations and discussions for these results?

**Questions:**

Please refer to Weaknesses.

---

> ### Author Response · Authors · 2025-11-22
>
> **Strengths:** Thanks for positive feedback on our motivation, graph design, and experiments. We address each weakness (W) below.
>
> **W1: How does the graph-driven branch actually influence the cache keys and final performance?**
>
> **Response:** We agree Eq. (4) needs clearer explanation. The cache keys and graph branch are jointly optimized: each key is initialized from the frozen CLIP feature, then its subspaces are refined by ISSA+FSP to produce $\tilde f_{train}$ for the second logit term. Training logits are formed as a residual sum of (i) a cache stream with $\Theta_{train}$ and (ii) a graph stream from ISSA+FSP$(f_{train})$. The loss on this combined output yields the gradient w.r.t. the keys:
> $\frac{\partial \mathcal{L}}{\partial \Theta_{train}} =
> \frac{\partial \mathcal{L}}{\partial \text{logits}} \cdot
> \frac{\partial z_{cache}}{\partial \Theta_{train}}.$
> Here, $\frac{\partial \mathcal{L}}{\partial \text{logits}}$ depends on the sum
> $z_{cache} + z_{CLIP}(\tilde f_{train})$.
> As the graph-refined CLIP branch becomes more predictive, it reshapes $\frac{\partial \mathcal{L}}{\partial \text{logits}}$, and thus indirectly guides the cache keys to align with the decision boundary induced by the graph-refined representation. This is analogous to a residual knowledge-distillation setup where a strong teacher branch shapes the gradients on a student branch without explicit parameter sharing.
>
> Empirically, this coupling is essential: removing ISSA+FSP (reducing to Tip-Adapter-F) yields large 16-shot drops even though the cache design and loss stay unchanged, for e.g., Aircraft 47.2→35.6, UCF101 84.2→78.0, Flowers102 97.9→94.8. We will clarify that (i) the graph branch is parameterized only through $\tilde f_{train}$, (ii) $\Theta_{train}$ is updated solely via the cache term, and (iii) ISSA+FSP affect the keys only through the shared loss, not direct feature substitution.
>
> **W2: why neighborhood statistics yield robust representations in ISSA?**
>
> **Response:** In ISSA, the “neighborhood’’ means other subspaces of the same CLIP feature. In our fully connected graph, each subspace sees all others and aggregates their information through statistical summaries (mean, max, std) with attention before merging this context into its embedding. Intuitively: Mean gives a background activation level, smoothing idiosyncratic spikes; Max highlights the strongest co-activations, capturing shared discriminative cues; Std. captures variability, helping separate stable patterns from noise. In low-shot regimes, individual dimensions are highly variable; aggregating neighbor statistics effectively denoises each subspace by anchoring it in the image’s global context. This is akin to statistical pooling/second-order methods in robust representation learning but adapted to subspace graphs.
>
> Empirically, attention over statistical nodes outperforms concatenation, projection, and simple pooling (Table 3), and removing intra-subspace edges (i.e., losing relational statistics) sharply reduces performance (e.g., Aircraft 16-shot: 47.2% with edges vs 37.9% without). We will make this robustness-via-context intuition clearer in revision and relate it to prior work on statistical graph aggregation.
>
> **W3: Outdated benchmarks; compare with recent ICML/ICCV/ICLR 2025.**
>
> **Response:** We focus on adapter-based rather than prompt-based methods; prompt tuning adds trainable layers and follows different adaptation protocols. For fairness, we compare only methods using the standard k-shot setting. Our submission already includes recent adapter baselines: GraphAdapter (NeurIPS’23), CLAP (CVPR’24), CLIP-Adapter (IJCV’24) across 11 datasets. We will add the ICCV 2025 adapter method [r3] and update related work.
>
> [r3] “Causal Disentanglement and Adapter Alignment for Few-Shot Recognition.” ICCV 2025.
>
> **W4: Interpretation of Table 6: do ~1 inter-slice cosine similarities mean redundancy or harm CLIP’s vision–language alignment?**
>
> **Response:** Table 6 reports average cosine across subspace pairs. Values near 1 mean ISSA+FSP makes slices agree on the same image-level semantics, not that features across different images collapse. High cosine indicates subspaces lie in a tight cone; their norms and higher-order structures still differ, and they are concatenated to form $\tilde f_{train}$, preserving fine variation. This reflects a shift from anisotropy to coherent semantics. Before refinement, inter-slice similarity is near zero/negative indicating arbitrary CLIP partitions disagree in direction. The graph encourages subspaces supporting the same label to co-activate. In few-shot settings, such redundancy is beneficial as mutually reinforcing subspaces stabilize predictions under limited data.
>
> CLIP’s vision–language alignment remains intact: we never modify the CLIP encoder or text classifier. Inference uses original zero-shot logits plus the residual cache term, so the graph only distills relational cues into cache keys without altering alignment.

---

> > ### Author Response · Authors · 2025-12-01
> >
> > Thank you for the detailed feedback. We trust that our rebuttal resolved the points you raised. We understand that the OpenReview security disruption may have restricted your ability to revise the rating during the final phase.

---

### Official Review · Reviewer_eVfs · 2025-11-01

**Soundness:** 3
**Presentation:** 3
**Contribution:** 3
**Rating:** 6
**Confidence:** 4

**Summary:**

This paper proposes a novel framework for few-shot adaptation of CLIP, building upon the "cache model" paradigm (e.g., Tip-Adapter-F). The authors argue that existing methods treat CLIP's global feature embedding as a monolithic, unstructured vector, thereby ignoring the rich relational information within the feature's dimensions. To address this, the proposed method introduces a "graph-driven cache refinement" process that is active only during training. The result is a method that (in theory) distills the rich, intra-feature relational knowledge from the graph into the cache keys, while maintaining the high efficiency (no GNN, no extra parameters) of the cache model at inference time. The authors show SOTA performance on 11 benchmarks, especially in the 1-8 shot regime.

**Strengths:**

Excellent Motivation and Core Idea: The paper's key insight—that a global CLIP embedding is not a monolithic vector but a structured representation whose internal subspaces can be refined—is excellent. The idea of using a GNN to model intra-feature relationships, rather than the typical inter-sample (transductive) relationships, is novel and clever.

Strong "Best of Both Worlds" Design: The framework's design is its strongest point. It intelligently "distills" the knowledge from the complex graph refinement process into the simple cache keys. This allows the model to leverage deep, relational feature learning during training while paying zero computational overhead for it at inference time. This is a very elegant solution to the usual trade-off between performance (adapters, prompts) and efficiency (cache models).

Comprehensive Empirical Validation: The paper presents SOTA results across 11 diverse datasets (Table 1), with particularly large gains in the low-shot (1-4) regime (e.g., +5% avg. at 1-shot over the next-best adapter). This is a very strong empirical contribution.

**Weaknesses:**

My concerns are minor and are mostly related to clarifying the exact mechanism of ISSA and the training.ISSA Aggregation (Eq. 1):

1.  The formulation of the ISSA aggregation in Eq. 1 is confusing and seems non-standard. It appears to define $H_u$ as a $3 \times F$ matrix of the three statistical aggregations (mean, max, std), and then applies a full self-attention mechanism within this tiny $3 \times 3$ "graph" of statistics.

Q1: Is this interpretation correct? If so, what is the intuition behind applying self-attention to the statistics themselves? A more standard GNN approach would be to concatenate the statistics ($[h_{mean}, h_{max}, h_{std}]$) into a single $1 \times 3F$ vector and apply a linear layer. The ablation in Table 3 shows "attn" is best, but the justification for why it's better than concatenation ("cat") or projection ("proj") is missing.

2. Training Procedure (Eq. 4): There is a potential ambiguity in the training loss (Eq. 4). The first term (cache retrieval) uses the original feature $f_{train}$, while the second term (CLIP classification) uses the graph-refined feature $\tilde{f}_{train}$.

Q2: Does the gradient from the $\tilde{f}_{train}$ term flow back to update $\Theta_{train}$? The text says: "This residual-style formulation ensures that gradients from the CLIP classifier stream guide the adaptation of cache keys". This implies $\tilde{f}_{train}$'s loss also updates $\Theta_{train}$, but the equation logits = ... + \tilde{f}_{train}W_c^T does not contain $\Theta_{train}$.

Please clarify the exact gradient flow. Is $\Theta_{train}$ only updated by the $f_{train}$ term? Or is $\Theta_{train}$ updated by both terms? Or, as a third possibility, is the refined feature $\tilde{f}_{train}$ used to also calculate the cache retrieval term (i.e., replacing $f_{train}$ in the first term)? This last option seems most logical but is not what Eq. 4 says.

3. Partitioning Strategy (Table 6): The authors claim that "contiguous" partitioning is better than "random," but the results in Table 6 show a negligible difference (e.g., 47.2 vs 47.0; 97.9 vs 97.8). This slightly undermines the motivation that the contiguous chunks have some inherent structure. Suggestion: The authors should tone down this claim. The real takeaway from this ablation seems to be that the method is robust to the partitioning strategy, as long as all feature dimensions are covered. The strong performance of both contiguous and random splits suggests the model is truly learning the relationships, rather than exploiting a pre-existing spatial structure in the feature vector.

**Questions:**

None

---

> ### Author Response · Authors · 2025-11-22
>
> **Strengths:** Thank you for the positive assessment. We appreciate your recognition of our subspace-refinement idea and the “best of both worlds” design that distills graph reasoning into an inference-free cache. Below, we address each weakness (W) and will retain this clarity in the revision.
>
> **W1: Clarifying exact mechanism of ISSA formulation in Eq. 1 and how we apply attention over mean/max/std statistics.**
>
> **Response:** Thanks for pointing this confusion. For each subspace we first compute three neighborhood statistics (mean, max, std) and stack them along a separate (first) dimension as a $3 \times F$ matrix $H_u$, to treat each row of $H_u$ as an individual statistical node rather than collapsing them into a single $1 \times 3F$ vector. We then apply a shared self-attention layer (with shared $W_Q$, $W_K$, $W_V$) over these three nodes and sum the self-attended outputs to obtain a single $1 \times F$ vector for that subspace. This design lets the model adaptively reweight and mix the complementary statistics, instead of relying on a fixed combination. In contrast, the “cat’’ variant in Table. 3 simply concatenates $[h_{\text{mean}}, h_{\text{max}}, h_{\text{std}}]$ into a $1 \times 3F$ vector, while the “proj’’ variant applies an additional linear layer to this concatenated vector to project it back to $1 \times F$, exactly as you described. Our attention-based design yields slightly but consistently perform better.  **Unlike “cat’’ or “proj”, which rely on a fixed, data-independent mixing of the statistics, our attention mechanism learns content-dependent weights, allowing each subspace to emphasize different cues depending on the image content. We will update the text around Eq.~1 to clarify this and highlight how ISSA differs from the “cat’’ and “proj’’ baselines in revised version.**
>
>
> **W2: Training procedure, gradient flow in Eq. 4, and clarification of the ambiguous sentence.**
>
> **Response:** Thanks for pointing this ambiguity. In our implementation, the cache keys and the graph branch are **jointly optimized, not independent.** Each cache key is initialized with the frozen CLIP feature of a support image; the same feature is partitioned into subspaces and passed through ISSA+FSP to produce the refined embedding $\tilde f_{\text{train}}$ used in the second term of the logits (Eq. 4). We form the training logits as a residual sum of two streams: (i) a cache stream that uses the keys $\Theta_{\text{train}}$ directly, and (ii) a graph stream that applies ISSA+FSP to $f_{\text{train}}$ and then the frozen CLIP textual classifier. The loss is applied to this combined prediction, so the gradient with respect to $\Theta_{\text{train}}$ comes from the cache term, but its magnitude and direction are shaped by how well the residual (cache + graph) logits match the label. In this sense, the graph-refined classifier stream acts as a teacher: when it produces a more confident or structured prediction, the shared loss encourages the cache logits to align with it, thus guiding the adaptation of the cache keys indirectly through the residual formulation rather than via a direct derivative through $\tilde f_{\text{train}}$. After training, the graph parameters are discarded, but the refined keys retain this distilled knowledge and are used for retrieval. **We will revise our inaccurate wording to “gradients from the graph classifier stream guide the adaptation of cache keys’’ to correctly reflect this teacher–student effect and clarify in Sec. 3 that the CLIP backbone and text encoder remain frozen throughout.**
>
>
> **W3: Clarification of contiguous vs random partitioning strategy and interpretation of table 6**
>
> **Response:** We agree that our wording around Table 6 can be toned down. Our intention was not to claim that contiguous slices encode strong pre-existing semantic structure, but to introduce a simple inductive bias aligned with the common interpretation of CLIP/ViT features (e.g., local groups of channels). Contiguous and random splits yield very similar performance (e.g., 47.2 vs 47.0 on Aircraft, 97.9 vs 97.8 on Flowers), which indicates that the graph is learning relationships between channel groups regardless of how they are assigned, as long as all dimensions are covered. We keep the contiguous partition as the default only because it is better and simpler to implement, but we will revise the text to emphasize robustness rather than strong prior structure in the feature ordering, aligning with your suggestion.

---

> > ### Author Response · Authors · 2025-12-01
> >
> > Thank you for your valuable feedback and thoughtful review. We trust that our rebuttal resolved the points you raised. We understand that the OpenReview security disruption may have restricted your ability to revise the rating during the final phase.

---

### Official Review · Reviewer_1zWJ · 2025-11-02

**Soundness:** 3
**Presentation:** 3
**Contribution:** 2
**Rating:** 4
**Confidence:** 4

**Summary:**

This paper improves few-shot image classification by enhancing CLIP’s task adaptability while keeping inference efficient. The authors introduce a training-only graph-based refinement framework that strengthens CLIP’s representations without adding test-time cost. It operates via two steps: subspace-level statistical aggregation to capture robust intra-sample dependencies, and subspace propagation to enrich contextual information. The refined cache keys are used at inference, but the graph module is discarded. Experiments across multiple benchmarks show consistent gains over CLIP-based methods and competitive state-of-the-art performance.

**Strengths:**

1. Proposes a subspace-level refinement strategy for CLIP-based few-shot learning, offering a different angle compared to prompt/adaptor tuning.

2. Training-only graph refinement is a practical design that improves representations without adding inference cost.

3. Method is clearly presented and experimentally validated across multiple benchmarks, showing consistent improvements.

**Weaknesses:**

1. The introduction quickly jumps into the technical proposal without clearly framing the core challenge, limitations of CLIP, and intuition behind subspace graph reasoning.

2. Although the empirical section is quite thorough, some ablations do not cover multiple datasets, slightly limiting conclusions about generality. More detailed computation-vs-gain analysis would strengthen claims about efficiency and practicality.

3. Missing comparisons with very recent (2024–2025) CLIP-adaptation or FSL works, especially those exploring subspace modeling or cache refinement approaches. Authors should add the latest baselines if possible.

4. While attractive, the paper could better articulate why training-time subspace propagation leads to robust inference-time performance without graphs, e.g., theoretical perspective or more analytical visualization.

**Questions:**

1. Could the authors more explicitly describe why subspace partitioning addresses CLIP’s bias in few-shot regimes? Any theoretical or empirical intuition beyond empirical gains?

2. Please further explain why the propagated subspace enhancement can be discarded at inference with minimal loss. Are there scenarios where this fails (e.g., domain shift, higher-shot scenarios)?

3. Can you provide a more detailed computational breakdown (training time, memory, inference speed) relative to prompt tuning and adapter baselines?

4. Are there 2025 FSL methods the authors can compare against, such as the latest CLIP-tuning or cache-refinement works?

5. Some ablations appear limited to a subset of datasets. Any results across additional benchmarks to verify robustness?

6. How sensitive is performance to the number of subspaces and graph connectivity?

---

> ### Author Response · Authors · 2025-11-22
>
> **Strengths:** Thanks for your positive remarks on our subspace refinement idea, the training-only graph design, the overall clarity and empirical support. We address each weakness (W) and question (Q) below.
>
> **W1 (Q1 & Q2): Motivation, CLIP bias, and intuition behind subspace graph**
>
> **Response:** We agree the Introduction can better highlight the core challenge and intuition. We will clarify that: (i) characterize CLIP’s few-shot bias as the dominance of coarse, pre-training directions and prompt sensitivity that suppress fine-grained cues and hurt performance on shifted domains (e.g., Aircraft, EuroSAT); (ii) standard cache methods treat the CLIP embedding as a single monolithic vector and thus cannot reweight or disentangle these factors; and (iii) motivate subspace graph reasoning to refine CLIP’s global vector representation by modeling intra-feature structure that exchanges information and re-balances salient dimensions before caching.
> Existing cache approaches treat the CLIP vector as a single unit, ignoring semantic structure across channel groups. Our subspace partitioning with ISSA+FSP explicitly models this internal structure. **Prior work in metric learning [r1] and self-supervised learning [r2] shows that deep features form semantic channel groups and that their interactions improve representation quality.** Motivated by this, we split CLIP features into channel groups and use ISSA+FSP to exchange local statistics and complementary cues among nodes. This lightweight refinement strengthens discriminative groups, suppresses noise-dominated ones, and effectively “opens’’ the embedding to correct CLIP’s few-shot bias. The recombined feature yields tighter, better-separated clusters (t-SNE on 5 datasets) and consistent gains. Table 6/8 shows raw subspaces have near-zero or negative similarity, but after refinement they align (≈0.99), showing the graph regularizes CLIP’s anisotropic embedding into a coherent, task-adapted geometry. We will move this intuition earlier and tie it directly to these diagnostics.
> Our graph is used only during training to learn better cache keys. At inference, predictions rely solely on cosine similarity between the query and keys, plus CLIP’s zero-shot logits; no graph mechanism is required.
>
> [r1] “Deep Metric Learning via Group Channel-Wise Ensemble.” Knowledge-Based Systems, 2023.
>
> [r2] “Self-Supervised Learning: Disentangled Group Representation as Feature.”, NeurIPS, 2021.
>
>
> **W2 (Q3, Q5 & Q6): Ablations, generality, computation vs gain, and sensitivity**
>
> **Response:** Thanks for this comment. Our method preserves the core benefit of cache-based designs: **fast, lightweight inference identical to Tip-Adapter-F**, since the graph is discarded after training. On ImageNet with ViT-B/16 and 8 subspaces, the graph adds only ~0.037M params and 1.12×10⁻³ GFLOPs (total ~0.86M params, 2.75×10⁻³ GFLOPs). In a 16-shot setup (512-d, batch 256, A40 GPU), memory increases minimally (2932→2940 MB, ~0.3%) and training time rises from 6→13 s/epoch, while inference cost remains unchanged. We will include a small table summarizing this computation-vs-gain trade-off.
> To assess sensitivity to subspace partitions (Q6), we ablated on Aircraft, Flowers102, and UCF101. Table 2 shows consistent trends: ViT-B/16 gives the strongest 16-shot performance (47.2%, 97.9%, 84.4%), and ∣S∣ = 8 is consistently best. Accuracy remains high across a wide range of ∣S∣ values (Aircraft 46.5–45.8%, Flowers 97.6–97.8%, UCF101 84.4–84.2%), demonstrating that our refinement is inherently stable and does not depend on a fragile partition size. To address Q5, we will include more dataset ablations to further strengthen generality.
>
> **W3 (Q4): Comparisons with very recent 2024–2025 methods**
>
> **Response:** Thanks for emphasizing this. We focus on adapter-based methods rather than prompt-based ones, since prompt tuning adds trainable layers and follows different protocols. For fairness, we compare only methods using the standard k-shot setting. To the best of our knowledge, our submission already includes recent adapter baselines (GraphAdapter NeurIPS’23, CLAP CVPR’24, CLIP-Adapter IJCV’24) across 11 datasets. We will also add the newly published ICCV 2025 adapter method [r3] and update the related-work discussion accordingly.
>
> [r3] “Causal Disentanglement and Adapter Alignment for Few-Shot Recognition.” ICCV 2025.
>
> **W4: Explanation of training-time propagation vs inference-time benefits**
>
> **Response:** Thanks for this comment. During training, subspaces exchange information via ISSA+FSP, acting as a regularizer that smooths noise, reinforces consistent cues, and improves feature geometry. The refined representations are stored into the cache, so this structure is fully distilled into the keys, and no graph is needed at inference. Fig. 3 (t-SNE) shows tighter, better-separated clusters after refinement, illustrating how training-time propagation improves inference-time retrieval.

---

> ### Author Response · Authors · 2025-11-27
>
> We expanded our analysis to UCF101, EuroSAT, and DTD (along with Aircraft and Flowers102) to address Q3, Q5, and Q6 and demonstrate greater generalizability. These results appear below and will be added to the revised paper and supplementary material.
>
> > **Accuracy (%) with varying CLIP visual encoder backbones (|S| = 8).**
> | **Dataset** | **RN50** | **RN101** | **ViT-B/32** | **ViT-B/16** |
> |-------------|----------|-----------|--------------|--------------|
> | **EuroSAT** | 78.1 | 79.0 | 76.5 | **87.2** |
> | **DTD**     | 65.8 | 67.3 | 65.8 | **73.4** |
>
> >**Accuracy (%) with varying number of graph nodes |S| using ViT-B/16 backbone (16-shot).**
> | **Dataset** | **S=4** | **S=8** | **S=16** | **S=32** | **S=64** |
> |-------------|---------|---------|----------|----------|----------|
> | **EuroSAT** | 83.6 | **87.2** | 84.2 | 76.3 | 76.2 |
> | **DTD**     | 69.6 | **73.4** | 70.0 | 69.0 | 68.8 |
>
> >**Impact of Teleport Probability γ on Accuracy (%) for 16-shot.**
> | **Dataset** | **0.1** | **0.3** | **0.5** | **0.7** | **0.9** |
> |-------------|-------------|---------|---------|---------|---------|
> | **UCF**      | **84.4** | 84.3 | 84.1 | 84.2 | 84.1 |
> | **EuroSAT**  | **87.2** | 87.0 | 85.9 | 85.4 | 84.9 |
> | **DTD**      | **73.4** | 70.3 | 70.0 | 70.0 | 69.5 |
>
> >**Impact of Propagation Steps T on Accuracy (%) for 16-shot.**
> | **Dataset** | **1** | **3** | **5** | **7** | **9** |
> |-------------|-----------|-------|-------|-------|-------|
> | **UCF**      | **84.4** | 84.2 | 84.0 | 84.0 | 83.8 |
> | **EuroSAT**  | **87.2** | 85.1 | 83.9 | 82.8 | 82.2 |
> | **DTD**      | **73.4** | 73.0 | 73.0 | 72.6 | 71.9 |
>
> >**Impact of Aggregation Mode on Accuracy (%) for 16-shot.**
> | **Dataset** | **attn** | **cat** | **proj** | **mean** | **max** | **sum** |
> |-------------|----------|----------|-----------|-----------|----------|----------|
> | **UCF**      | **84.4** | 84.1 | 84.0 | 83.8 | 83.5 | 83.8 |
> | **EuroSAT**  | **87.2** | **87.2** | 87.0 | 86.3 | 85.7 | 85.5 |
> | **DTD**      | **73.4** | 73.1 | 72.8 | 70.0 | 69.7 | 69.5 |
>
> >**Accuracy (%) of ISSA and FSP across K-shot settings for EuroSAT and DTD.**
> | **Dataset** | **ISSA** | **FSP** | **K=16** | **K=8** | **K=4** | **K=2** | **K=1** |
> |-------------|----------|---------|----------|---------|---------|---------|---------|
> | **EuroSAT** | ✗ | ✗ | 84.5 | 77.9 | 74.1 | 66.1 | 59.5 |
> |             | ✗ | ✓ | 79.1 | 78.5 | 76.3 | 66.2 | 60.0 |
> |             | ✓ | ✗ | 86.9 | 81.9 | 78.1 | 67.4 | 63.7 |
> |             | ✓ | ✓ | **87.2** | **81.9** | **78.1** | **67.4** | **63.7** |
> | **DTD**     | ✗ | ✗ | 66.5 | 62.7 | 57.4 | 53.7 | 49.6 |
> |             | ✗ | ✓ | 70.6 | 66.3 | 59.7 | 53.5 | 50.9 |
> |             | ✓ | ✗ | 70.9 | 66.3 | 59.1 | 53.5 | 51.1 |
> |             | ✓ | ✓ | **73.4** | **68.9** | **62.3** | **55.3** | **52.4** |
>
> >**Impact of modeling relations between subspaces (with edges vs. without) for EuroSAT and DTD.**
> | **Shot** | **Edges** | **EuroSAT** | **DTD** |
> |----------|-----------|-------------|----------|
> | **1-shot** | w/o | 54.7 | 50.1 |
> |            | w   | **63.7** | **52.4** |
> | **4-shot** | w/o | 61.4 | 51.6 |
> |            | w   | **78.1** | **62.3** |
> | **16-shot** | w/o | 68.7 | 58.7 |
> |             | w   | **87.2** | **73.4** |
>
> >**Computational efficiency of ours vs. Tip-Adapter-F on ImageNet 16-shot.**
> | **Method**       | **Acc. (%)** | **Param. (M)** | **G-FLOPs**      | **Peak Memory (MB)** | **Train-time/epoch (sec)** |
> |------------------|--------------|------------|-------------------|------------------------|-----------------------------|
> | Tip-Adapter-F    | 65.5         | 0.82      | 1.64 × 10⁻³       | 2932                   | 6                           |
> | Ours             | 73.1         | 0.86      | 2.75 × 10⁻³       | 2940                   | 13                          |
>
>
>
> >**Few-shot performance (%) across varying shots on EuroSAT and DTD, with extended 32- and 64-shot results.**
> | **Dataset** | **1-shot** | **2-shot** | **4-shot** | **8-shot** | **16-shot** | **32-shot** | **64-shot** |
> |-------------|------------|------------|------------|------------|-------------|-------------|-------------|
> | **EuroSAT** | 63.7 | 67.4 | 78.1 | 81.9 | 87.2 | 87.6 | **89.4** |
> | **DTD**     | 52.4 | 55.3 | 62.3 | 68.9 | 73.4 | 73.2 | **75.8** |

---

> > ### Comment · Reviewer_1zWJ · 2025-11-27
> >
> > The rebuttal satisfactorily addresses my main concerns with clearer motivation, expanded ablations, and additional comparisons. While some presentation aspects (e.g., figure quality) still leave room for improvement, the overall contribution is now stronger. I am therefore raising my score.

---

> > > ### Author Response · Authors · 2025-11-28
> > >
> > > Thank you for the positive follow-up and for raising your score. We will regenerate all figures at higher resolution with improved contrast and layout in the updated version of the paper.

---

### Author Response · Authors · 2025-11-22
**Final Comments from Authors**

We thank reviewers **1zWJ**, **eVfs**, and **JNSy** for their thorough evaluations, constructive feedback, and positive comments on our submission. We are grateful to reviewer **1zWJ** for increasing the rating following our rebuttal.
For reviewers **eVfs** and **JNSy**, we trust that our rebuttal resolved the points you raised. We understand that the OpenReview security disruption may have restricted your ability to revise the rating during the final phase of the review process.


**Final thoughts from authors to ACs/Reviewers:** Our paper introduces a subspace-level refinement paradigm for CLIP-based few-shot learning: instead of treating CLIP’s global embedding as a monolithic vector (typical in cache models) or modifying CLIP via prompts/adapters (typical in tuning methods), we **partition the embedding into subspaces and learn intra-feature relationships** with a lightweight graph module at training time. This yields a best-of-both-worlds design: **relational refinement during training, but zero added inference cost** because the learned improvements are distilled into the cache keys and inference remains a simple cache + zero-shot CLIP combination. Reviewers consistently highlighted (i) the novelty of modeling intra-feature structure (not inter-sample transductive graphs), (ii) the practicality of **training-only** refinement, and (iii) strong, consistent gains across **11 benchmarks**, especially in the 1–4 shot regime.

**Clarifications and rebuttal actions (addressing reviewer weaknesses):** The main concerns raised are largely about exposition and causal pathways, not empirical validity. We address them as follows:

-	**Clearer framing & intuition (CLIP bias → subspaces → graphs).** We revise the Introduction to explicitly state the core challenge: in few-shot regimes, CLIP features can be anisotropic and overly dominated by coarse, pretraining-induced directions. Subspace partitioning provides multiple complementary “cues” of the embedding; graph reasoning encourages cross-subspace consistency and stabilizes the representation used to form cache keys.

-	**How the graph influences cache-key optimization (Eq. (4) clarification).** We clarify that the graph-refined branch shapes cache learning through the shared loss: even if cache keys appear only in the cache-logit term, the gradient signal driving the key updates depends on the sum of cache logits and graph-refined CLIP logits. Thus, stronger/refined CLIP logits reshape the loss landscape and steer cache keys toward better decision boundaries, despite the graph not being used at inference.

-	**Why refinement can be discarded at inference.** We make explicit that the graph’s role is to produce better cache keys during training; once learned, inference depends only on frozen CLIP + the refined cache. We add an ablation supporting the “distillation into cache” interpretation, and we discuss potential edge cases (e.g., extreme domain shift) transparently.

-	**Partitioning strategy claim (contiguous vs random).** We agree with the reviewer: Table 6 shows only negligible differences. We tone down any claim that contiguous partitions encode inherent semantic structure and instead highlight the stronger takeaway: robustness to partitioning strategy (coverage matters more than ordering), which actually strengthens the method’s generality.

-	**Inter-slice cosine similarity near 1 (redundancy / alignment concerns).** We clarify that high average cosine after refinement indicates reduced anisotropy and improved coherence, not collapse to identical vectors. Importantly, CLIP encoders and text prototypes remain frozen, and the zero-shot CLIP stream is preserved at inference; empirically there is no evidence of degraded vision–language alignment, as performance improves consistently over baselines.

-	**Generality + efficiency reporting.** We extend key ablations to additional datasets (beyond the subset used initially). We also provide a more explicit computation-vs-gain breakdown (training time, memory, and inference speed), emphasizing that our inference cost matches cache baselines by design.

-	**Recent (2024–2025) comparisons.** We clarified that our paper already includes recent adapter baselines (GraphAdapter NeurIPS’23, CLAP CVPR’24, CLIP-Adapter IJCV’24) across 11 datasets. We have also added the newly published ICCV 2025 adapter method and update the related-work discussion accordingly.

Reviewers view the core idea as novel, elegant, and practical: graph-based relational learning inside the embedding with no inference burden, backed by strong multi-benchmark results. The raised weaknesses primarily reflect places where we must better explain the optimization coupling, interpretability of subspace behavior, and broaden reporting; we addressed these with clearer exposition, strengthened ablations, and added compute analysis without altering the method’s central claims.

---

### Note · Program_Chairs · 2026-01-17
**Submission Desk Rejected by Program Chairs**

The following references in this submission do not refer to real documents and/or have major errors in bibliographic information:

 Imtiaz Ziko, Gauthier Gidel, Karim Louedec, Eugene Belilovsky, and Ioannis Mitliagkas. Laplacianshot: Laplacian-based representation for few-shot classification. In Proceedings of the IEEE/CVF Conference on Computer Vision and Pattern Recognition, pp. 6548-6557, 2020.
Yu-Jie Liu, Bernt Schiele, and Yi Sun. Transductive propagation network for few-shot learning. In International Conference on Learning Representations, 2019.